# Improved Expressivity of Hypergraph Neural Networks through High-Dimensional Generalized Weisfeiler-Leman Algorithms

**Detian Zhang** [1]  **Chengqiang Zhang** [1]  **Yanghui Rao** [2]  **Qing Li** [3]  **Chunjiang Zhu** [4]

## Abstract

The isomorphism problem is a key challenge in both graph and hypergraph domains, crucial for applications like protein design, chemical pathways, and community detection. Hypergraph isomorphism, which models high-order relationships in real-world scenarios, remains underexplored compared to the graph isomorphism. Current algorithms for hypergraphs, like the 1-dimensional generalized Weisfeiler-Lehman test (1-GWL), lag behind advancements in graph isomorphism tests, limiting most hypergraph neural networks to 1-GWL's expressive power. To address this, we propose the high-dimensional GWL (k-GWL), generalizing k-WL from graphs to hypergraphs. We prove that k-GWL reduces to k-WL for simple graphs, and thus develop a unified isomorphism method for both graphs and hypergraphs. We also successfully establish a clear and complete understanding of the GWL hierarchy of expressivity, showing that (k+1)-GWL is more expressive than k-GWL with illustrative examples. Based on k-GWL, we develop a hypergraph neural network model named k-HNN with improved expressive power of k-GWL, which achieves superior performance on real-world datasets, including a 6% accuracy improvement on the Steam-Player dataset over the runner-up. Our code is available at https://github.com/talence-zcq/KGWL.

[1]School of Computer Science and Technology, Soochow University, Suzhou, China [2]School of Computer Science and Engineering, Sun Yat-sen University, Guangzhou, China [3]Department of Computing, the Hong Kong Polytechnic University, Hong Kong [4]Department of Computer Science, University of North Carolina at Greensboro, Greensboro, NC. Correspondence to: Chunjiang Zhu <c_zhu3@uncg.edu>.

*Proceedings of the 42$^{nd}$ International Conference on Machine Learning*, Vancouver, Canada. PMLR 267, 2025. Copyright 2025 by the author(s).

## 1. Introduction

Hypergraphs have gained increasing attention in recent years, because of their ability to naturally model a wide range of systems where high-order relationships exist among their constituting parts. In hypergraphs, a hyperedge allows the connection of an arbitrary number of nodes and can easily abstract high-order relationships in social systems (Lotito et al., 2022), biology (Sanchez-Gorostiaga et al., 2019), co-authorship collaboration networks (Wu et al., 2022), ecology (Grilli et al., 2017), and neuroscience (Xiao et al., 2019). Despite the powerful expressiveness, hypergraph research, such as representation learning (Antelmi et al., 2023) and isomorphism test (Feng et al., 2024), has been underexplored compared to the graph counterpart because of the inherent complexity and the lack of popular tools. In addition, simply transforming hypergraphs to simple graphs either leads to an inevitable loss of information (Zhou et al., 2006) or adds significantly many auxiliary nodes/edges that increase space and time requirements in downstream tasks (Yang et al., 2019; Yoon et al., 2020). It is demanding to develop hypergraph isomorphism test and representation learning algorithms. The two topics are closely related as the expressivity of hypergraph neural networks (HNNs), i.e., what functions the HNN models can approximate, can resort to the power of isomorphism test in distinguishing non-isomorphic hypergraphs (Morris et al., 2023).

In hypergraph representation learning, HGNN (Feng et al., 2019) used the clique expansion technique to approximate hypergraphs as graphs, and then ran graph convolution (Kipf & Welling, 2016) but cannot recover hypergraph structures with hyperedges in another hyperedge. HyperGCN (Yadati et al., 2019) replaced each hyperedge with an incomplete clique. Then HyperSAGE (Arya et al., 2020) first learned to embed hypergraphs directly by propagating messages in a two-stage procedure, which can be described as passing messages from vertices to hyperedges, and then back to vertices from hyperedges. UniGNN (Huang & Yang, 2021) unified the two-stage hypergraph learning by generalizing several Graph Neural Networks (GNNs) to HNNs, pointing out that its expressive power is bounded by the 1-dimensional generalized Weisfeiler-Leman test (1-GWL). Subsequent works, such as AllSetTransformer (Chien et al.,

2022) and ED-HNN (Wang et al., 2023), are also of the two-stage HNN type. Feng et al. (2024) further introduced a hypergraph isomorphism test algorithm, referred to as 1-GWL. It also mentioned that 1-GWL can degrade into 1-WL when applied to graphs, thus promoting the theoretical unification of isomorphism tests.

Both GNNs and 1-dimensional WL test (1-WL) (Weisfeiler & Leman, 1968) aggregate messages from neighbors and iteratively update node representations and can be considered as neural and discrete variants. In the graph domain, researchers have striven for a good understanding that classic GNNs, including GCN and GIN (Xu et al., 2018), have expressive power upper bounded by 1-WL (Morris et al., 2019), and a series of subgraph GNNs (Qian et al., 2022; Frasca et al., 2022; Zhang et al., 2023a) have expressive power at most 3-WL. Here, the high-dimensional WL test (k-WL) (Immerman & Lander, 1990) can be viewed as using k-tuples of vertices to replace the original vertices in the graph for isomorphism test. However, currently there is **no research study that can enhance the expressivity of 1-GWL in a systematic way as k-WL, resulting in that the expressive power of most current HNNs is limited to 1-GWL**. Therefore, a significant research gap exists in extending the 1-GWL algorithm to its higher-dimensional counterparts and designing HNNs guided by these more powerful algorithms.

In this paper, we propose a higher-dimensional version of 1-GWL, namely k-GWL. It is a generalization of k-WL that can accept hypergraphs as input and process complex and diverse high-order relations therein. Based on the different aggregation methods of k-tuple neighbors, we propose two variants, namely the folklore variant k-GFWL and the oblivious variant k-GOWL. We prove that the proposed k-GWL can degenerate to k-WL when applied to simple graphs, *subject to* that the initialization of k-tuple features uses our sub-hypergraph extraction method that deletes singleton hyperedges (Theorem 5.1). It unifies the high-dimensional isomorphism tests for both simple graphs and hypergraphs. Moreover, we prove the expressive power of (k+1)-GWL surpasses that of k-GWL for $k \geq 1$ (Theorems 5.2 and 5.3) with illustrative examples. The (k+1)-GOWL has the same expressive power as k-KFWL for $k \geq 2$ (Theorem 5.4). With these, we successfully establish a clear and complete understanding of the GWL expressivity hierarchy (Eq. (7)).

Furthermore, we develop high-dimensional HNN models, namely k-HNNs, based on the proposed k-GWL. We show that k-HNNs are provably more powerful than existing HNN models and have expressive power at most k-GWL (Theorems 6.1 and 6.2). In our experiments, k-HNNs achieve superior performance across all datasets. Notably, on the Steam-Player dataset, our methods outperform the second-best approach by about 6% in accuracy.

## 2. Related Work

### 2.1. Expressivity of GNNs

Two seminal works (Xu et al., 2018; Morris et al., 2019) related the problems of graph isomorphism test and GNNs, revealing that any GNN architecture cannot be more powerful than 1-WL in terms of distinguishing non-isomorphic graphs. Morris et al. (2019) proposed a new GNN architecture to overcome the limitations of 1-WL by learning features over the k-tuple vertices instead of vertices. Later, several works gradually proposed the incorporation of unique vertex identifiers (Vignac et al., 2020), randomized vertex labels (Sato et al., 2021; Abboud et al., 2021) and subgraph information (Zhao et al., 2021; Zhang & Li, 2021; Papp et al., 2021; Papp & Wattenhofer, 2022; Bouritsas et al., 2022; Qian et al., 2022; Frasca et al., 2022) into the vertex attributes to further enhance the expressiveness of GNNs. Zhou et al. (2023) unified this framework to propose the (k,$l$)-WL algorithm, which can be seen as applying k-WL to a graph with $l$ vertices that are assigned additional features. They also provided a theoretical analysis of the expressivity of (k,$l$)-WL, establishing an expressivity hierarchy for algorithms with different values of k and $l$. Zhang et al. (2023b) studied the expressivity related to biconnectivity and designed a 1-WL's variant that can encode general distance metrics. Bodnar et al. (2021) developed a WL for handling simplicial complexes (generalizations of graphs). A more comprehensive discussion of the WL test in GNNs can be found in the survey (Morris et al., 2023).

### 2.2. HNNs and Their Expressivity

The development of HNNs shifted from single-stage message-passing framework (Feng et al., 2019; Yadati et al., 2019) to a two-stage update/aggregation process (Arya et al., 2020; Huang & Yang, 2021; Chien et al., 2022; Wang et al., 2023). In particular, Chien et al. (2022) unified a whole class of two-stage models with multiset functions, which allows to learn the most adequate update/aggregation for individual dataset/task. Inspired by hypergraph diffusion (Li et al., 2020), Wang et al. (2023) proposed ED-HNN that computes specialized hyperedge-to-node messages for nodes in a hyperedge and can approximate any permutation-equivariant hyperedge diffusion. Huang & Yang (2021) developed a unified framework for GNNs and HNNs. It also proposed 1-GWL for hypergraph isomorphism test following (Böker, 2019) and bounded the expressive power of UniGNN by 1-GWL. Feng et al. (2024) separately developed 1-GWL and also designed hypergraph subtree and hyperedge kernel methods that improved earlier hypergraph kernels (Wachman & Khardon, 2007; Bai et al., 2014). In this paper, we propose a k-dimensional generalized WL algorithm (k-GWL), filling the research gap in hypergraphs where no higher expressive-power algorithm existed. Additionally,

we design k-HNNs based on k-GWL, with its theoretical expressivity upper bounded by k-GWL. Luo et al. (2023) studied the expressivity of hypergraph neural networks and constructed hierarchies of arity (the maximum number of vertices in hyperedges) and depth. For example, when the depth is larger than a certain value, a neural logic machine with a larger arity is more expressive. In contrast, this paper generalizes k-WL to k-GWL, unifies graph and hypergraph isomorphism tests, and establishes a clear generalized WL hierarchy for hypergraphs.

## 3. Preliminary on the WL Algorithm

Let $\mathcal{HG} = (\mathcal{V}, \mathcal{E}, \mathcal{X})$ be a hypergraph, where $\mathcal{V}$ is the set of vertices, $\mathcal{E}$ is the set of hyperedges, and $\mathcal{X}$ is the vertex feature vectors. In the incidence matrix $\mathcal{H} \in \{0,1\}^{|\mathcal{V}| \times |\mathcal{E}|}$, each entry $\mathcal{H}(v, e)$ indicates whether vertex $v$ is in the hyperedge $e$. $\mathcal{N}_e(v)$ denotes the set of hyperedge neighbors of vertex $v$, i.e., hyperedges containing $v$, and $\mathcal{N}_v(e)$ denotes the set of vertex neighbors of hyperedge $e$, i.e., vertices in $e$.

The **1-dimensional Weisfeiler-Leman (1-WL)** (Weisfeiler & Leman, 1968) is a classic algorithm for graph isomorphism test and has expressivity no worse than classic GNNs. Let $\mathcal{G}$ be a graph with possibly labeled vertices. In iteration 0, the coloring $c_1^{(0)}(v)$ is initialized by the label of vertex $v$ or a uniform value if no label is provided. 1-WL iteratively updates the color for vertex $v$ according to its original color and the colors of its neighbors. Specifically, the color is computed as

$$c_1^{(t)}(v) = \mathrm{HASH}\big(c_1^{(t-1)}(v), \{\!\{c_1^{(t-1)}(u) | u \in \mathcal{N}(v)\}\!\}\big)$$

where HASH bijectively maps the above pair to a unique color which has not been used in previous iterations.

It is well known that 1-WL is not able to distinguish all non-isomorphic graphs (Cai et al., 1992). The **k-dimensional Weisfeiler-Leman (k-WL)** was proposed to improve the expressiveness (Immerman & Lander, 1990). It assigns colors to k-tuples of vertices and iteratively updates them. In iteration 0, the color $c_k^{(0)}(\mathbf{v})$ of k-tuple $\mathbf{v} \in \mathcal{G}^k$ is initialized as the isomorphism/atomic type of the subgraph induced by k vertices in $\mathbf{v}$. In iteration $t > 0$, the color of tuple $\mathbf{v}$ is computed by

$$
\begin{aligned}
c_k^{(t)}(\mathbf{v}) = \mathrm{HASH}\big(&c_k^{(t-1)}(\mathbf{v}), \\
&\{\!\{\{\theta_j(\mathbf{v}, w) \,|\, w \in \mathcal{V}(\mathcal{G})\} \,|\, 1 \le j \le k\}\!\}\big)
\end{aligned}
\tag{1}
$$

where $\theta_j(\mathbf{v}, w)$ refers to a high-order neighbor of $\mathbf{v}$ obtained by replacing the $j^{th}$ element of $\mathbf{v}$ by $w$. There is another variant of k-WL that differs slightly in how they aggregate neighborhood information and is denoted *folklore k-WL (k-FWL)* in machine learning literature (Maron et al., 2019; Morris et al., 2019). The above one is called *oblivious k-WL (k-OWL)*. k-OWL aggregates the colors of neighbors obtained by putting all elements in one position, while k-FWL aggregates the colors of neighbors obtained by putting one element in all k positions. Grohe & Otto (2015) proved that k-OWL is as powerful as the folklore (k-1)-FWL for $k > 2$.

However, the above isomorphism algorithms cannot be directly applied to hypergraphs. Recently, the **1-dimensional Generalized WL (1-GWL)** was proposed for the hypergraph isomorphism test (Feng et al., 2024; Huang & Yang, 2021). 1-GWL considers two neighborhoods: vertex's hyperedge neighbors $\mathcal{N}_e(v)$ and hyperedge's vertex neighbors $\mathcal{N}_v(e)$. In iteration 0, the colors of $c_1^{(0)}(e)$ and $c_1^{(0)}(v)$ are initialized using their original labels, if any. In iteration $t > 0$, the color updating rules are as follows.

$$c_1^{(t)}(e) = \mathrm{HASH}\big(c_1^{(t-1)}(e), \{\!\{c_1^{(t-1)}(u) \mid u \in \mathcal{N}_v(e)\}\!\}\big)$$

$$c_1^{(t)}(v) = \mathrm{HASH}\big(c_1^{(t-1)}(v), \{\!\{c_1^{(t)}(u) \mid u \in \mathcal{N}_e(v)\}\!\}\big)$$

Huang & Yang (2021) proved that the two-stage HNN, UniGNN, has expressivity upper bounded by 1-GWL. It is easy to see this also holds for AllSet and thus many HNNs that are its special case (Thm. 3.4 in (Chien et al., 2022)).

## 4. High-Dimensional Generalized WL

In this section, we present our main method k-GWL for the hypergraph isomorphism test. For clarity, tuples are ordered and allow repetitive elements; multisets are unordered and allow repetitive elements; and sets are unordered and do not allow repetitive elements. In k-GWL, we consider k-tuples, which allow repetitions of vertices. There are two types of hypergraphs: the input hypergraph and the k-tuple hypergraph (see its construction in Section 4.1.2). Hyperedges in both types of hypergraphs are sets, that is unordered.

### 4.1. How k-GWL Works

For our proposed k-GWL hypergraph isomorphism test, the k-WL algorithm is a natural reference. The principle of k-WL is to color k-tuples instead of single vertices. To achieve that, it builds a *k-tuple graph* with each vertex as a k-tuple, initializes the k-tuple features, and then applies 1-WL graph coloring in the k-tuple graph. To develop the k-GWL hypergraph isomorphism test algorithm, we ask the following research questions (RQs):

**RQ1**: How can we initialize the k-tuple features in the k-tuple hypergraph while ensuring that, when given a simple graph input, the k-tuple features keep the same as k-WL?

**RQ2**: How can we construct the k-tuple hypergraph from the original hypergraph so that we can apply 1-GWL hypergraph coloring?

**RQ3**: How to develop HNNs with an expressive power up-

per bounded by k-GWL, significantly improving the upper bound 1-GWL of most existing HNNs?

In the following, we discuss how to address these research questions one-by-one (with RQ3 answered in Section 6) and explain our k-GWL methods.

### 4.1.1. k-TUPLE FEATURE INITIALIZATION (RQ1)

The initialization of k-tuple features is a crucial step in realizing the strong expressive power of k-dimensional hypergraph coloring. A random initialization cannot capture unique characteristics of the collection of k vertices in each k-tuple and is limited in getting theoretical guarantees on the expressivity. A more desirable way is to initialize k-tuple features according to the *isomorphism type* of the *induced sub-hypergraph*. In this way, two k-tuples get the *same* initial color if their induced sub-hypergraphs are isomorphic. This helps build necessary connections among those k-tuples to ensure the theoretical expressivitiy. We formally define two isomorphic hypergraphs:

**Definition 4.1.** Given hypergraphs $\mathcal{HG}_1 = (V_1, H_1, X_1)$, $\mathcal{HG}_2 = (V_2, H_2, X_2)$ with vertex features $X_1$, $X_2$ and k-tuples $\mathbf{s}^1$, $\mathbf{s}^2$ in $\mathcal{HG}_1$ and $\mathcal{HG}_2$ (with the $i^{th}$ element as $\mathbf{s}_i^1$, $\mathbf{s}_i^2$), respectively. We say $\mathbf{s}^1$, $\mathbf{s}^2$ are isomorphic and have *the same isomorphism type (or called atomic type)* iff

- $\forall i_1, i_2 \in [k], \mathbf{s}_{i_1}^1 = \mathbf{s}_{i_2}^1 \leftrightarrow \mathbf{s}_{i_1}^2 = \mathbf{s}_{i_2}^2$.

- $\forall i \in [k], X_{1,\mathbf{s}_i^1} = X_{2,\mathbf{s}_i^2}$.

- $\forall i_1, ..., i_n \in [k], (\mathbf{s}_{i_1}^1, ..., \mathbf{s}_{i_n}^1) \in H^1 \leftrightarrow (\mathbf{s}_{i_1}^2, ..., \mathbf{s}_{i_n}^2) \in H^2$.

Extracting the sub-hypergraph induced by (vertices in) a k-tuple is a required and crucial step, before computing its isomorphic type. While computing the subgraph induced by a k-tuple in a simple graph is trivial (obtained by retaining only the rows and columns of the adjacency matrix corresponding to vertices in the k-tuple), the counterpart operation in hypergraphs needs special and careful considerations. Taking Fig. 1 as an example, the top graph and hypergraph are equivalent and we are given 2-tuples $(v_1, v_3)$ and $(v_1, v_2)$. But if we simply keep the corresponding rows and non-empty columns in the incidence matrix (similar to simple graphs), then the resulting sub-hypergraph in the bottom right would not be equivalent as the subgraph in the bottom left! This is because after the extraction, there are multiple hyperedges that contain only a single vertex. This would prevent the degeneration of k-GWL to k-WL for input simple graphs.

To address this issue, we propose to *delete singleton hyperedges* (those with only a single vertex). With this extra step, the subgraph and sub-hypergraph extractions become consistent and aligned for input simple graphs, paving the road for subsequent theoretical analysis. The singleton deletions are reasonable, since the main role of hyperedge connections is to represent higher-order relations among multiple vertices. With only one vertex in a hyperedge, it might be good to remove it. We put detailed explanation in Appendix C. In our experiments, we find that the removal of singleton hyperedges significantly enhances the hypergraph classification performance.

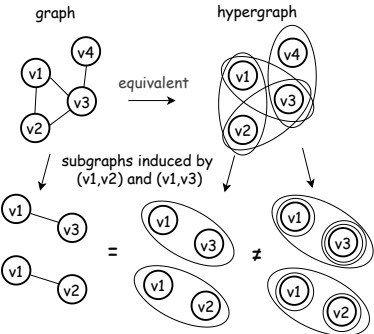

*Figure 1.* Induced sub-hypergraph extraction.

Formally, as the first step of k-GWL, we initialize the color of k-tuples $\mathbf{s} = (v_1, ..., v_k) \in \mathcal{V}(\mathcal{HG})^k$ instead of individual vertices.

$$c_k^{(0)}(\mathbf{s}) = \text{ISO}(\mathcal{HG}_{\text{sub}}(\mathbf{s})) \tag{2}$$

where $\mathcal{HG}_{\text{sub}}(\mathbf{s})$ represents the sub-hypergraph of $\mathcal{HG}$ induced by the k-tuple $\mathbf{s}$ using the above technique, and the ISO function colors a sub-hypergraph according to its isomorphism type.

### 4.1.2. k-TUPLE HYPERGRAPH CONSTRUCTION (RQ2)

In k-WL, a k-tuple graph is constructed where two k-tuples are connected, called neighbors, if they differ only by one of the k entries. Then, 1-WL can be applied on the graph to lift the expressivity up to k-WL. In the hypergraph setup, 1-GWL must be applied in hypergraphs, but it is not clear how to construct hypergraph structures based on k-tuples to preserve their high-order relations.

In the proposed k-GWL, the connection patterns between k-tuples are still rooted at and managed by their neighborhood relations, i.e., being the same in all k entries but only one entry. First, let $\theta_i(\mathbf{s}, w)$ be the k-tuple obtained by replacing the $i^{th}$ element of $\mathbf{s}$ by $w$. We define the *neighbors* of a *k-tuple* $\mathbf{s} = (v_1, ..., v_k)$ in $\mathcal{V}(\mathcal{HG})^k$ as k-tuples obtained by replacing the $j$-th component of $\mathbf{s}$ by $w \in \mathcal{V}(\mathcal{HG})$. The definition of the neighbors of a k-tuple $N(\mathbf{s})$ is the same as the graph setting.

$$N(\mathbf{s}) = \{\theta_j(\mathbf{s}, w) = (v_1, ..., v_{j-1}, w, v_{j+1}, ..., v_k) \mid 1 \leq j \leq k, w \in \mathcal{V}(\mathcal{HG})\} \tag{3}$$

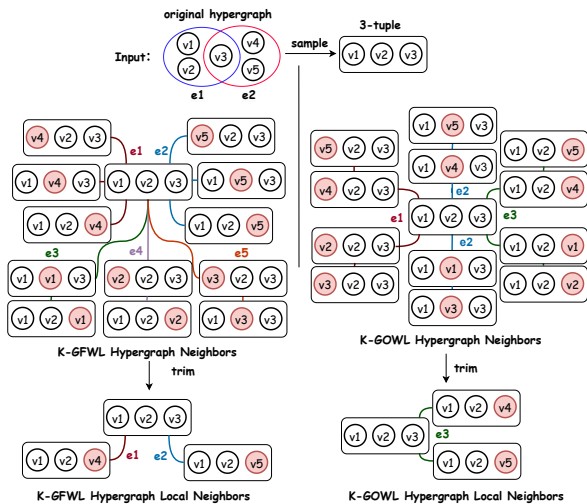

*Figure 2.* Illustration of k-GFWL and k-GOWL hypergraph neighbors and hypergraph local neighbors for k-tuple $(v_1, v_2, v_3)$, where $k = 3$. When considering set-based k-tuples, k-tuples with repeated vertices are absent, such as $e_3$, $e_4$, and $e_5$ in the left.

It is natural to connect a k-tuple and *each* of its neighbors using a hyperedge, similar to k-WL. But this would result in $k * n$ hyperedges only for a k-tuple, making it prohibitively large and limiting the scalability. Instead, it is reasonable to connect a k-tuple with *multiple* k-tuples in its neighbors using a hyperedge, since they formulate a high-order neighborhood relationship. We construct two types of $k$-dimensional hypergraphs, depending on changing position $j$ or vertex $w$ to identify the neighboring k-tuples (which will affect the aggregation methods in applying 1-GWL later). They are the $k$-dimensional *folklore* hypergraph $\mathcal{HG}_{k,f}$ and $k$-dimensional *oblivious* hypergraph $\mathcal{HG}_{k,o}$ for k-tuples. The k-GWL variants running on them are called k-dimensional *folklore* GWL (k-GFWL) and k-dimensional *oblivious* GWL (k-GOWL), respectively.

In the folklore hypergraph $\mathcal{HG}_{k,f}$, a k-tuple **s** has $n$ hyperedges, one for each vertex $w \in \mathcal{V}(\mathcal{HG})$ represented as $\big(\mathbf{s}, \theta_1(\mathbf{s}, w), \theta_2(\mathbf{s}, w), \cdots, \theta_k(\mathbf{s}, w)\big)$. The set of hyperedges $\mathcal{E}_{k,f}$ among k-tuples in $\mathcal{HG}_{k,f}$ is defined by the following formula:

$$\mathcal{E}_{k,f} = \{\!\{ \big(\mathbf{s}, \theta_1(\mathbf{s}, w), \theta_2(\mathbf{s}, w), \cdots, \theta_k(\mathbf{s}, w)\big) \mid \\ w \in \mathcal{V}(\mathcal{HG}), \, \mathbf{s} \in \mathcal{V}(\mathcal{HG})^k \}\!\} \quad (4)$$

For example, in Fig. 2 left, we consider the 3-tuple $(v_1, v_2, v_3)$ as our k-tuple **s** with $k = 3$. If we choose $v_4$ as vertex $w$ to replace a vertex at each position of **s**, the resulting 3-tuples $(v_4, v_2, v_3)$, $(v_1, v_4, v_3)$, $(v_1, v_2, v_4)$ together with **s** itself are included in hyperedge $e_1$. If we choose $v_5$ as vertex $w$, the 3-tuples $(v_5, v_2, v_3)$, $(v_1, v_5, v_3)$, $(v_1, v_2, v_5)$ and **s** itself are included in hyperedge $e_2$. If $v_1$ is chosen, the 3-tuples $(v_1, v_1, v_3)$, $(v_1, v_2, v_1)$ and **s** itself

are included in hyperedge $e_3$.

In the oblivious hypergraph $\mathcal{HG}_{k,o}$, a k-tuple **s** has $k$ hyperedges, one for each position $j$ represented as $(\{\mathbf{s}\} \cup \{\theta_j(\mathbf{s}, w) \mid w \in \mathcal{V}(\mathcal{HG})\})$. The set of hyperedges $\mathcal{E}_{k,o}$ in the oblivious hypergraph $\mathcal{HG}_{k,o}$ is then defined as follows:

$$\mathcal{E}_{k,o} = \{\!\{ \big(\{\mathbf{s}\} \cup \{\theta_j(\mathbf{s}, w) \mid w \in \mathcal{V}(\mathcal{HG})\}\big) \mid \\ 1 \le j \le k, \, \mathbf{s} \in \mathcal{V}(\mathcal{HG})^k \}\!\} \quad (5)$$

As an example, in Fig. 2 right, when we choose the first position as $j$ to replace every vertex with $v_1$, the resulting 3-tuples $(v_2, v_2, v_3)$, $(v_3, v_2, v_3)$, $(v_4, v_2, v_3)$, and $(v_5, v_2, v_3)$ together with **s** are placed in hyperedge $e_1$. When we choose the second and third positions as $j$ respectively, we have that $(v_1, v_1, v_3)$, $(v_1, v_3, v_3)$, $(v_1, v_4, v_3)$, $(v_1, v_5, v_3)$ and **s** are in hyperedge $e_2$ and $(v_1, v_2, v_1)$, $(v_1, v_2, v_2)$, $(v_1, v_2, v_4)$, $(v_1, v_2, v_5)$ and **s** are in hyperedge $e_3$. It should be noted that the construction of k-tuple hypergraphs in k-GWL does not depend on the actual hyperedges in the original hypergraph, which are only used in the isomorphism type. It will be different for HNNs in Section 6 though.

Both k-GWL and k-WL allow repetitions of vertices in k-tuples. In the initialization of k-tuple features, they extract sub-hypergraphs and subgraphs induced by the set of vertices in k-tuples and then use the isomorphism type as features, where the set operation before extraction removes vertex repetitions. For the construction of k-tuple hypergraphs and k-tuple graphs, because tuples allow repetitions, the vertex replacement strategies in the oblivious and folklore variants ("all elements in one position" vs. "one element in all k positions") still work with no issues.

### 4.1.3. APPLYING 1-GWL IN K-TUPLE HYPERGRAPH

With a k-tuple hypergraph $\mathcal{HG}_k$ (either $\mathcal{HG}_{k,f}$ or $\mathcal{HG}_{k,o}$) at hand, we can proceed with applying 1-GWL to iteratively update the color of every k-tuple **s**, $c_k^{(t)}(\mathbf{s})$ at iteration $t$. In each iteration, 1-GWL updates both colors of hyperedges $c_k^{(t)}(\mathbf{e})$ and k-tuples $c_k^{(t)}(\mathbf{s})$ in the k-tuple hypergraph $\mathcal{HG}_k$. The initial colors $c_k^{(0)}(\mathbf{e})$ for all hyperedges **e** are the same while the initial k-tuple colors $c_k^{(0)}(\mathbf{s})$ are set based on their isomorphism type. Let $\mathcal{N}_\mathbf{s}(\mathbf{e})$ be k-tuple neighbors of hyperedge **e** and $\mathcal{N}_\mathbf{e}(\mathbf{s})$ be hyperedge neighbors of k-tuple **s**. The updating rules are defined as follows:

$$c_k^{(t)}(\mathbf{e}) = \text{HASH}\big(c_k^{(t-1)}(\mathbf{e}), \{\!\{ c_k^{(t-1)}(\mathbf{s}) \mid \mathbf{s} \in \mathcal{N}_\mathbf{s}(\mathbf{e}) \}\!\}\big)$$
$$c_k^{(t)}(\mathbf{s}) = \text{HASH}\big(c_k^{(t-1)}(\mathbf{s}), \{\!\{ c_k^{(t)}(\mathbf{e}) \mid \mathbf{e} \in \mathcal{N}_\mathbf{e}(\mathbf{s}) \}\!\}\big)$$

The process can be considered as two-stage updating: first update hyperedge colors by the colors of k-tuples in it and its color in the previous iteration, and then update k-tuple

colors by the newly updated colors of hyperedges containing it and its color in the previous iteration. For example, in Fig. 2 left, we compute the color of the 3-tuple $(v_1, v_2, v_3)$ as follows. We first aggregate the colors of $(v_4, v_2, v_3)$, $(v_1, v_4, v_3)$, $(v_1, v_2, v_4)$, $(v_1, v_2, v_3)$, and $e_1$ to update the color of $e_1$. Similarly, the color of $e_2$ is updated. Then, we use the updated colors of $e_1$ and $e_2$ and the previous color of $(v_1, v_2, v_3)$ to update the 3-tuple's color. After each iteration, we compare the multisets of k-tuple colors for two input hypergraphs. We can determine whether they are isomorphic or not in at most $h$ iterations. The pseudocode of proposed k-GWL is shown in Appendix B and multiple complete executions of k-GWL are shown in Appendix D.

### 4.2. Complexity

The computational complexity of the proposed k-GWL with $h$ iterations is $\mathcal{O}(hm)$, where $m$ is the complexity in one iteration and $m = |\mathcal{V}|\overline{d}_{\mathbf{v}} + |\mathcal{E}|\overline{d}_{\mathbf{e}}$. Here $\overline{d}_{\mathbf{v}}$ and $\overline{d}_{\mathbf{e}}$ are the average degrees of vertices and hyperedges in the k-tuple hypergraph $\mathcal{HG}_k$ (instead of the original hypergraph). In $\mathcal{HG}_{k,f}$ and $\mathcal{HG}_{k,o}$, we can derive the number of hyperedges $|\mathcal{E}|$ based on their definitions: $|\mathcal{E}_{k,f}|$ in $\mathcal{HG}_{k,f}$ is $n^k \cdot n$, while in $\mathcal{HG}_{k,o}$, $|\mathcal{E}_{k,o}|$ is $n^k \cdot k$. Here $n^k$ represents the total number of k-tuples in $\mathcal{V}(\mathcal{HG})$. Additionally, the average degree $\overline{d}_{\mathbf{e}}$ in $\mathcal{HG}_{k,f}$ is $k$, whereas in $\mathcal{HG}_{k,o}$, it is $n$. Therefore, the total time complexity for both k-GFWL and k-GOWL algorithms are $O(h \cdot n^k \cdot n \cdot k) = O(hk \cdot n^{k+1})$. Here the run-time is independent of the number of hyperedges in the original hypergraph.

## 5. Theoretical Properties

In this part, we discuss theoretical results on the expressive power of the two variants of k-GWL to distinguish non-isomorphic hypergraphs. All the proofs are given in Appendix A. For two isomorphism algorithms $A$ and $B$, we denote their respective final colors of hypergraph $\mathcal{HG}$ as $c_A(\mathcal{HG})$ and $c_B(\mathcal{HG})$. We say:

- $A$ is **more powerful** than $B$ ($B \preceq A$) if for any pair of hypergraphs $\mathcal{HG}_1$ and $\mathcal{HG}_2$, $c_A(\mathcal{HG}_1) = c_A(\mathcal{HG}_2) \Rightarrow c_B(\mathcal{HG}_1) = c_B(\mathcal{HG}_2)$. Otherwise, there exists a pair of hypergraphs that $B$ can distinguish while $A$ cannot, denoted as $B \not\preceq A$.

- $A$ is **as powerful as** $B$ ($A \cong B$) if $B \preceq A \wedge A \preceq B$.

- $A$ is **strictly more powerful** than $B$ ($B \prec A$) if $B \preceq A \wedge A \not\cong B$.

### 5.1. The GWL Hierarchy and Connection with Existing WL algorithms

We first show that k-GWL degenerates to k-WL for simple graphs (instead of hypergraphs), unifying high-dimensional

isomorphism tests for both simple graphs and hypergraphs.

**Theorem 5.1.** *Given a simple graph* $\mathcal{G} = \{V, E\}$*, let* $c_{(k,wl)}(\mathbf{s})$ *and* $c_{(k,gwl)}(\mathbf{s})$ *be unique color labels that the k-WL and k-GWL assign the k-tuple* $\mathbf{s}$ *to, respectively. There exists a bijective function* $\phi$ *that maps the color* $c_{(k,wl)}(\mathbf{s})$ *from k-WL to k-tuple color* $c_{(k,gwl)}(\mathbf{s})$ *from k-GWL.*

To construct the GWL hierarchy, our theoretical results indicate that as the dimension k increases, the expressive power of k-GWL *strictly* increases and (k+1)-GOWL has the same expressivity as k-GFWL for $k \geq 2$. We will show shortly that 2-GOWL has a stronger expressive power than 1-GFWL.

**Theorem 5.2.** $\forall k \geq 1$, *(k+1)-GOWL* $\succ$ *k-GOWL.*

**Theorem 5.3.** $\forall k \geq 1$, *(k+1)-GFWL* $\succ$ *k-GFWL.*

**Theorem 5.4.** $\forall k \geq 2$, *(k+1)-GOWL* $\cong$ *k-GFWL.*

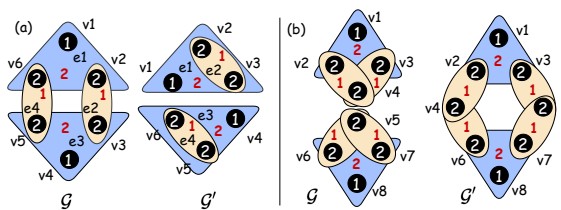

*Figure 3.* (a) A Hard Instance that 1-GWL fails but 2-GOWL succeeds; (b) A Harder Instance that 1-GWL and 2-GOWL fail but 2-GFWL succeeds.

In the graph realm, the WL hierarchy is quite clear that

$$\begin{aligned} 1\text{-OWL} &\cong 1\text{-FWL} \cong 2\text{-OWL} \\ &\prec 3\text{-OWL} \cong 2\text{-FWL} \qquad (6) \\ &\prec 4\text{-OWL} \cong 3\text{-FWL} \cdots \end{aligned}$$

and most GNNs (such as subgraph GNNs (Frasca et al., 2022)) have expressive power no greater than 3-OWL. The progress in the realm of hypergraphs lags much behind, with only 1-GWL developed before our work. In this work, *we successfully establish a clear and complete understanding of the GWL hierarchy*, which is outlined below:

$$\begin{aligned} 1\text{-GOWL} &\cong 1\text{-GFWL} \prec 2\text{-GOWL} \\ &\prec 3\text{-GOWL} \cong 2\text{-GFWL} \qquad (7) \\ &\prec 4\text{-GOWL} \cong 3\text{-GFWL} \cdots \end{aligned}$$

Remarkably, unlike that 2-OWL $\cong$ 1-WL in the WL hierarchy, we have 2-GOWL $\succ$ 1-GWL in the hypergraph GWL hierarchy. To showcase, we give a hard pair of hypergraphs in Fig. 3(a) which 2-GOWL successfully distinguishes but 1-GOWL and 1-GFWL fail. We also provide another pair of hypergraphs in Fig. 3(b) to confirm 2-GFWL $\succ$ 2-GOWL. Both 1-GWL and 2-GOWL fail

to distinguish the non-isomorphic hypergraphs, whereas 2-GFWL successfully identifies them. We refer readers to Appendix D for detailed and step-by-step executions of k-GWL in these hard instances. In fact, the hypergraphs in Fig. 3(a) can be distinguished only by the initialization step of 2-GOWL, whereas those in Fig. 3(b) require the subsequent color refinement in 2-GFWL. The established GWL expressivity hierarchy can not only help improve our understanding on the theoretical capability of existing methods, but also guide the development of highly expressive and practical hypergraph models.

## 5.2. Further Improving HNN Expressivity by Adding Vertex Labels

Zhou et al. (2023) devised a fairly general method to enhance the GNN expressivity through adding limited labels to vertices. The method can be applied to our k-GWL to further enhance the expressivity. Their idea is to assign extra labels $1, \cdots, l$ to $l$ vertices, but runs k-WL on the whole graph with the additional labels only for those $l$ vertices. This is repeated for every possible labeled graph and then the final representation of the graph is aggregated from the representations of all labeled graphs. This produces (k,l)-WL with a strictly stronger expressive power than k-WL for $l \geq 2$. It is straightforward to implement the approach in k-GWL, which results in (k,l)-GWL with even higher expressivity. However, the heavy computational cost of $O(h \cdot \binom{n}{l}^l \cdot n^k \cdot n \cdot k)$ makes it impractical. We perform an additional experiment on (1,2)-GWL to validate the performance as shown in Appendix E.6. We leave the development of efficient sampling methods for hypergraphs and adding more expressive vertex labels (e.g., structural and positional encodings for vertices), instead of using only vertex IDs in $l$-tuples, for future work.

## 5.3. Applying k-WL to Graphs transformed from Hypergraphs

One could transform hypergraphs to graphs via a bijective mapping, such as the star expansion or line expansion (Yang et al., 2022), and then apply k-WL on the transformed graphs for isomorphism testing. This looks promising at first glance. However, even though the transformation is bijective, it does not guarantee the results of k-WL on the transformed graphs are the same as those of k-GWL on the original hypergraphs. For instance, we can find two non-isomorphic hypergraphs where k-GWL can distinguish them but k-WL cannot distinguish their transformed graphs. Figure 3(a) provides such an example: while 1-WL (equivalent to 2-OWL) fails to distinguish the transformed graphs, 2-GOWL successfully identifies the original hypergraphs. Therefore, this indirect method does not achieve the same expressive power as our k-GWL algorithm.

## 6. k-Dimensional Hypergraph Neural Networks (k-HNNs)

In the following, to address RQ3, we propose k-HNNs based on the k-GWL algorithm, whose expressive power is upper bounded by k-GWL. Due to scalability and limited GPU memory, we consider set-based k-tuples in the k-GWL, called k-sets, where ordering and repeated vertices in a tuple are ignored. We treat each possible k-set as a vertex and construct a hypergraph structure based on the k-sets. As discussed earlier, we can construct two types of k-set hypergraphs, $\mathcal{HG}_{k,f}$ and $\mathcal{HG}_{k,o}$, based on Equations (4) and (5), respectively. However, to further leverage the structural information in the original hypergraph $\mathcal{HG}$ and reduce GPU memory usage, we define a **local neighborhood** construction for the k-set hypergraph. The *local neighbors* of a *k-tuple* $\mathbf{s} = (v_1, ..., v_k)$ are defined as follows:

$$N(\mathbf{s}) = \{\theta_j(\mathbf{s}, w) = (v_1, ..., v_{j-1}, w, v_{j+1}, ..., v_k) \mid \\ 1 \leq j \leq k, w \in \mathcal{V}(\mathcal{HG}), (w, v_j) \in \mathcal{E}(\mathcal{HG})\} \quad (8)$$

In the pruned k-set hypergraph structures, $\mathcal{HG}_{k,f}^l$ and $\mathcal{HG}_{k,o}^l$, we trim the original neighbors of each k-set $\mathbf{s}$ by retaining only those neighbors with the differing nodes $v_j$ and $w$ belonging to the same hyperedge in the original hypergraph $\mathcal{HG}$. For example at the bottom of Fig. 2, the local neighbors of $(v_1, v_2, v_3)$ contain only 3-sets $(v_1, v_2, v_4)$ and $(v_1, v_2, v_5)$, with other 3-sets pruned such as $(v_4, v_2, v_3)$ and $(v_1, v_4, v_3)$. This is because, in the original hypergraph, only $v_3$ shares a hyperedge with $v_4$ and $v_5$ but $v_1, v_2$ do not. We observe subtle differences in the resulting local k-set hypergraph structures: in the k-GFWL version, $(v_1, v_2, v_4)$ and $(v_1, v_2, v_5)$ belong to two separate hyperedges, whereas in the k-GOWL version, they are in the same hyperedge. Unlike k-GWL, for k-HNNs based on local neighborhoods, the actual hyperedges of the input hypergraph do influence the construction of the k-set hypergraph.

Given the k-set hypergraph $\mathcal{HG}_k^l$ ($= \mathcal{HG}_{k,f}^l$ or $\mathcal{HG}_{k,o}^l$), we can apply any base HNN (of expressivity at most 1-GWL) in k-HNN to significantly enhance the expressivity to be at most k-GWL. Similar to k-GWL, we first initialize the isomorphism type of all k-sets based on their induced sub-hypergraphs. In the implementation, one could use 1-GWL to determine the isomorphism type, which shows promising empirical performance in our experiment since induced sub-hypergraphs are often small. To take advantages of the base HNN, we combine the embeddings learned by the base HNN with the isomorphism-type features $f^{iso}(\mathbf{s})$ to get the k-set initial features, which can be expressed as:

$$x_{\mathbf{s}}^{(0)} = \sigma([f^{iso}(\mathbf{s}), \sum_{v \in \mathbf{s}} x_v^{(T)}] \cdot W_0)) \quad (9)$$

where $x_{\mathbf{s}}^{(0)}$ is the initial feature of k-set $\mathbf{s}$ in $\mathcal{HG}_k^l$, $x_v^{(T)}$ is the embedding of vertex $v$ learned by the base HNN, and $W_0$

is the learnable parameters. We also define that all initial features of hyperedges in $\mathcal{HG}_k^l$ are the same.

Then in each k-HNN layer $t > 0$, we can compute the feature vector for each k-set $\mathbf{s}$ in $[\mathcal{V}(\mathcal{HG})]^k$ by the base HNN. In our implementation, we adopt the following variant of UniGNN as the base HNN:

$$
\begin{aligned}
x_{\mathbf{e}}^{(t)} &= \sigma(D_{\mathbf{e}}^{-1} H_k \cdot x_{\mathbf{s}}^{(t-1)} \cdot W_{\mathbf{e}}^{(t)}) + x_{\mathbf{e}}^{(t-1)} \\
x_{\mathbf{s}}^{(t)} &= \sigma(D_{\mathbf{s}}^{-1} H_k \cdot x_{\mathbf{e}}^{(t)} \cdot W_s^{(t)}) + x_{\mathbf{s}}^{(t-1)}
\end{aligned}
\tag{10}
$$

where $x_{\mathbf{e}}^{(t)}$ and $x_{\mathbf{s}}^{(t)}$ are the latent representations of hyperedge $\mathbf{e}$ and k-tuple $\mathbf{s}$ in $\mathcal{HG}_k^l$ at layer $t$, respectively. $W_{\mathbf{e}}^{(t)}$ and $W_{\mathbf{s}}^{(t)}$ are the learnable parameters. $\sigma$ is an activation function, such as ReLU. $H_k$ is the incidence matrix of hypergraph $\mathcal{HG}_k^l$. $D_{\mathbf{s}}$ and $D_{\mathbf{e}}$ are the diagonal matrices of the k-set degrees and hyperedge degrees. Additionally, we add residual connections (He et al., 2016) as a means of alleviating the oversmoothing phenomenon.

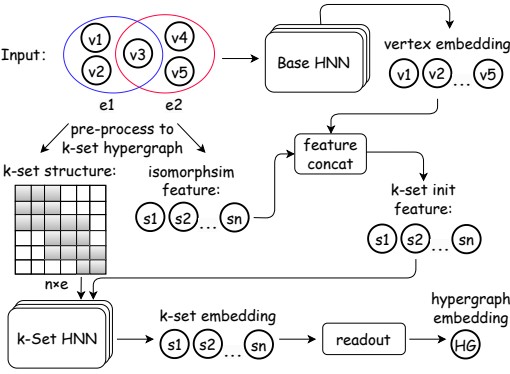

*Figure 4.* Illustration of the proposed k-dimensional HNNs.

After calculating the embeddings $x_s^{(T)}$ for all k-sets in the final layer ($T$-th layer), we can apply a permutation-invariant readout function, such as the sum function, in the k-set embeddings and obtain a hypergraph-level embedding as: $f(\mathcal{HG}) = \sum_{\mathbf{s} \in \mathcal{V}(\mathcal{HG})^k} x_{\mathbf{s}}^{(T)}$. This resulting hypergraph embedding can then be used for downstream tasks, such as hypergraph classification. The complete workflow of the model is shown in Fig. 4.

**Expressivity** The expressivity analysis of k-HNNs is based on k-tuples, instead of k-sets. To demonstrate the expressivity of k-HNNs, let $c \sqsubseteq d$ denote that k-tuple coloring $c$ refines k-tuple coloring $d$, or equivalently, $c(\mathbf{v}) = c(\mathbf{w})$ implies $d(\mathbf{v}) = d(\mathbf{w})$ for every k-tuple $\mathbf{v}, \mathbf{w}$. We prove that the coloring $c_k^{(t)}$ from k-GWL refines the features $x_{\mathbf{s}}^{(t)}$, which means that k-HNNs have expressive power upper bounded by k-GWL. We further show that when the aggregation, update, and readout functions are injective, the expressivitiy of k-HNNs is the same as k-GWL. The development of k-HNNs allows us to have a new way to design

hypergraph deep learning models with a target expressivity k-GWL, similar to the relation between k-IGNs and k-WL (Maron et al., 2019).

**Theorem 6.1.** *Given a hypergraph $\mathcal{HG} = \{\mathcal{V}, \mathcal{E}\}$ and let $k \geq 2$. Then for all iterations $t \geq 0$, for initial features $x_{\mathbf{s}}^{(0)}$ consistent with the initial colorings $c_k^{(0)}$ of k-GWL and for all weights $W^{(t)}$,*

$$
c_k^{(t)} \sqsubseteq x_{\mathbf{s}}^{(t)}
\tag{11}
$$

**Theorem 6.2.** *Let $\mathcal{A} : \mathcal{HG} \rightarrow \mathcal{R}^d$ be a k-HNN based on k-tuple neighborhood. With a sufficient number of layers, for any pair of hypergraphs that k-GWL determines as non-isomorphic, $\mathcal{A}$ also determines them as non-isomorphic if (1) the aggregation and update functions in the base HNN of $\mathcal{A}$ are injective, and (2) $\mathcal{A}$'s hypergraph embedding readout function is injective.*

# 7. Experiments

In this section, we give brief discussions on the experiments and the details of experimental setups and results are in Appendix E.

**Datasets:** In the experiments, we adopt three types of real-world hypergraph classification datasets proposed in HIC [14], which are IMDB, Steam-Player, and Twitter-Friend, ranging from movie staffs, game players, to social media. The IMDB dataset further contains four distinct subsets. To assess the efficacy of capturing distinct correlation structures, all datasets exclude the original vertex features. Keeping the original vertex features makes it easier to distinguish non-isomorphic hypergraphs, due to the potentially very different vertex features. Hence, it is a common practice to exclude these features and focus on structural information only. See statistics of the datasets and the constructed k-tuple hypergraphs in Tables 5, 6 and 7 of Appendix E.1.

**Compared Methods:** All experiments are conducted on a server with 1080Ti GPU 11GB, except for 3-HNNs running on a GPU cluster with 40GB memory. Due to the two different neighbor aggregation approaches, we obtain two variants of our models: k-FHNNs and k-OHNNs. For fair comparisons, we choose competitive hypergraph deep learning methods as baselines, including MLP, HyperGCN (Yadati et al., 2019), HNHN (Dong et al., 2020), UniGNNII (Huang & Yang, 2021), AllSetTransformer (Chien et al., 2022), ED-HNN (Wang et al., 2023), and HIC adapted from the 1-GWL subtree kernel method (Feng et al., 2024). The number of layers and the number of parameters for all the models are in Appendix E.7.

**Main Results** The experimental results for three types of real-world hypergraph datasets are presented in Table 1. First, our proposed two HNN models outperform other compared methods across all datasets. Notably, on the Steam-Player dataset, 2-FHNN outperforms the runner-up model,

*Table 1.* Experimental Results on Various Datasets

|  | IMDB-Dir-Form | IMDB-Dir-Genre | IMDB-Wri-Form | IMDB-Wri-Genre | Steam-Player | Twitter-Friend |
|---|---|---|---|---|---|---|
| MLP | 62.97±1.55 | 71.62±1.31 | 52.94±1.31 | 34.73±3.16 | 56.30±0.99 | 60.31±1.98 |
| HyperGCN | 65.70±3.12 | 78.46±1.37 | 49.72±4.34 | 31.03±5.71 | 59.02±1.27 | 63.92±2.11 |
| HNHN | 67.10±1.46 | 79.30±1.92 | 55.29±2.86 | 44.91±2.84 | 58.48±1.09 | 61.23±0.99 |
| UniGCNII | 64.90±1.57 | 75.16±0.92 | 52.41±3.10 | 40.19±2.08 | 58.89±1.40 | 60.92±2.32 |
| AllSetTransformer | 66.76±2.55 | 79.31±0.94 | 52.67±4.77 | 54.09±2.41 | 61.87±1.87 | 64.03±1.34 |
| ED-HNN | 66.18±1.73 | 74.65±1.81 | 50.07±3.13 | 35.07±2.10 | 58.35±2.22 | 60.76±2.18 |
| HIC | 66.19±1.49 | 79.43±0.63 | 49.74±5.09 | 49.83±3.35 | 58.35±1.01 | 62.37±1.61 |
| 2-OHNN | 67.25±2.35 | **79.75±1.14** | 55.35±3.74 | 50.08±1.89 | 65.97±1.37 | **64.12±1.30** |
| 2-FHNN | **68.11±2.46** | 78.52±1.07 | **55.36±3.30** | 45.44±1.24 | **67.53±1.23** | 62.37±1.91 |
| 3-OHNN (Sample Size 10) | 67.66±2.59 | 79.11±1.11 | 52.75±4.12 | **57.50±3.55** | 63.00±2.01 | 62.97±2.04 |
| 3-FHNN (Sample Size 10) | 67.07±2.33 | 78.90±1.10 | 50.56±2.69 | 53.84±2.05 | 61.50±1.59 | 63.05±3.67 |

*Table 2.* Results on Different Sub-Hypergraph Extraction Methods

|  | 2-FHNN | 2-FHNN-S | 2-OHNN | 2-OHNN-S |
|---|---|---|---|---|
| IMDB-Dir-Form | **68.11±2.4** | 58.55±1.5 | 67.25±2.3 | 60.11±2.0 |
| IMDB-Dir-Genre | 78.52±1.0 | 76.33±1.2 | **79.75±1.1** | 74.79±1.0 |
| IMDB-Wri-Form | **55.36±3.3** | 53.06±2.9 | 55.35±3.7 | 51.94±2.4 |
| IMDB-Wri-Genre | 45.44±1.2 | 36.38±1.5 | **50.08±1.8** | 35.95±2.3 |
| Steam-Player | **67.53±1.2** | 59.56±1.0 | 65.97±1.3 | 59.17±1.7 |
| Twitter-Friend | 62.75±2.1 | 60.89±1.6 | **64.12±1.3** | 61.23±2.1 |

*Table 3.* Results on Sample Vertex Numbers (2-FHNN)

|  | 5 | 10 | 15 | 20 | All |
|---|---|---|---|---|---|
| IMDB-Dir-Form | 66.45±1 | 65.97±2.5 | 66.51±1.4 | 66.67±2.8 | **68.11±2.5** |
| IMDB-Dir-Genre | 76.24±1 | 77.49±1.2 | 77.40±1.4 | 77.04±2.7 | **78.52±1.1** |
| IMDB-Wri-Form | 52.22±4 | 52.78±4.4 | 48.61±5.2 | 51.11±5.2 | **55.36±3.3** |
| IMDB-Wri-Genre | **47.93±3.3** | 43.10±2.3 | 38.36±2 | 37.76±1.6 | 45.44±1.2 |
| Steam-Player | 61.81±2.2 | 63.43±1.7 | 65.88±1.9 | 65.98±1.5 | **67.53±1.2** |
| Twitter-Friend | 62.15±1.3 | 61.08±1.2 | **62.85±2.1** | 62.31±1.8 | 62.75±2.2 |

*Table 4.* Time per Epoch (s) for Different Samplings (2-FHNN)

|  | 5 | 10 | 15 | 20 | All |
|---|---|---|---|---|---|
| IMDB-Dir-Form | 7.58 | 8.46 | 8.46 | 8.58 | 10.27 |
| IMDB-Dir-Genre | 13.81 | 14.59 | 15.34 | 15.54 | 20.71 |
| IMDB-Wri-Form | 1.49 | 1.57 | 1.59 | 1.59 | 1.72 |
| IMDB-Wri-Genre | 4.67 | 4.86 | 4.83 | 5.06 | 5.89 |
| Steam-Player | 10.75 | 11.10 | 11.21 | 10.87 | 11.16 |
| Twitter-Friend | 6.82 | 7.13 | 7.63 | 7.34 | 7.77 |

AllSetTransformer, by 6% in accuracy. This advantage stems from our model, guided by k-GWL, being able to better capture the high-order structures within hypergraph data. Second, we have run experiments on k-HNNs for $k = 3$ based on the vertex sampling strategy with sample size 10 in a GPU cluster with 40GB memory. Although not the full 3-HNNs without the sampling, 3-HNNs with the sampling have comparable performance to 2-HNNs across the datasets, while significantly outperforming in the IMDB-Wri-Genre dataset. Third, although 2-FHNN is theoretically more advantageous than 2-OHNN, experimental results reveal mixed performance across datasets, with each outperforming in half of the datasets.

We also conduct comparative experiments for two sub-hypergraph extraction methods. We append "-S" to the model name when keeping all hyperedges, including singleton hyperedges, in extracting sub-hypergraphs. Table 2 shows the proposed induced sub-hypergraph method not only supports the unification of graphs and hypergraphs, but also offers significant empirical advantages.

**Influence of the Vertex Sampling Strategy** In k-HNNs, the time complexity inevitably grows exponentially with the increase in k. To reduce the complexity of k-tuple hypergraph construction, we propose a vertex sampling strategy: we sort the vertex set of each hypergraph dataset by degree and select the top $m$ vertices with the highest degrees. These sampled vertices are then used to construct the k-tuple hypergraph and train the models, while the remaining vertices are ignored. Considering the distribution of vertex numbers across the datasets, we set $m$ to 5, 10, 15, and 20, resulting in four experimental groups. The selected results in Table 3 show the effectiveness of the sampling method in approximating the hypergraph classification accuracy.

**Runtime Results** To investigate the time complexity of the k-HNN model, we record the average time required to run one epoch across different datasets and different sampled vertices. Selected results are presented in Table 4. We observe that as the number of sampled vertices decreases, the time required for the model to complete one epoch also decreases. For instance, in the IMDB-Dir-Genre dataset, where the hyperedge degree is higher, the time to complete one epoch decreases by nearly 50% when going from full sampling to sampling 5 vertices.

## 8. Conclusion

We establish a generalized WL hierarchy for hypergraphs with increasing expressivity, with the notable difference between 1-GFWL vs. 2-GOWL, unlike its graph counterpart. The hierarchy allows us to design hypergraph neural networks with the desired expressivity, improves upon most existing hypergraph neural networks whose expressive power is upper bounded by 1-GWL. It is interesting to develop computationally efficient hypergraph deep learning models with provably high expressivity for a large value of k in k-GWL. It is also an interesting direction to characterize the expressive power of graph and hypergraph transformers.

## Acknowledgments

This work was supported by Project Funded by the Priority Academic Program Development of Jiangsu Higher Education Institutions (PAPD). The work of Yanghui Rao was supported by the National Natural Science Foundation of China (62372483). Qing Li has been supported by the Hong Kong Research Grants Council under General Research Fund (project no. 15200023) as well as Research Impact Fund (project no. R1015-23). Chunjiang Zhu has been supported by Natural Science Foundation CCF-2349369.

## Impact Statement

This paper establishes a clear and complete understanding on the expressivity hierarchy of GWL for hypergraphs, which guides the development of highly expressive, practical hypergraph models. The hypergraph theory and general hypergraph deep learning models proposed in this paper do not have special societal impacts that should be emphasized.

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

# A. Proof

For two isomorphism algorithms $A$ and $B$, we denote their respective final colors of hypergraph $\mathcal{HG}$ as $c_A(\mathcal{HG})$ and $c_B(\mathcal{HG})$. We say:

- $A$ is **more powerful** than $B$ ($B \preceq A$) if for any pair of hypergraphs $\mathcal{HG}_1$ and $\mathcal{HG}_2$, $c_A(\mathcal{HG}_1) = c_A(\mathcal{HG}_2) \Rightarrow c_B(\mathcal{HG}_1) = c_B(\mathcal{HG}_2)$. Otherwise, there exists a pair of hypergraphs that $B$ can distinguish while $A$ cannot, denoted as $B \not\preceq A$.

- $A$ is **as powerful as** $B$ ($A \cong B$) if $B \preceq A \wedge A \preceq B$.

- $A$ is **strictly more powerful** than $B$ ($B \prec A$) if $B \preceq A \wedge A \not\cong B$. For example, for any pair of hypergraphs $\mathcal{HG}_1$ and $\mathcal{HG}_2$, $c_A(\mathcal{HG}_1) = c_A(\mathcal{HG}_2) \Rightarrow c_B(\mathcal{HG}_1) = c_B(\mathcal{HG}_2)$, and there exists at least one pair of hypergraphs $\mathcal{HG}_1$ and $\mathcal{HG}_2$ such that $c_B(\mathcal{HG}_1) = c_B(\mathcal{HG}_2)$, but $c_A(\mathcal{HG}_1) \neq c_A(\mathcal{HG}_2)$.

**Theorem A.1.** *(Theorem 5.1 in the main text) Given a simple graph $\mathcal{G} = \{V, E\}$, let $c_{(k,wl)}(\mathbf{s})$ and $c_{(k,gwl)}(\mathbf{s})$ be unique color labels that the k-WL and k-GWL assign the k-tuple $\mathbf{s}$ to, respectively. There exists a bijective function $\phi$ that maps the color $c_{(k,wl)}(\mathbf{s})$ from k-WL to k-tuple color $c_{(k,gwl)}(\mathbf{s})$ from k-GWL.*

*Proof.* We only prove the oblivious variants on k-OWL and k-GOWL and the proofs for the folklore variants follow analogously.

**Proof of injection:** We first construct a function $\phi := c_{(k,wl)}(\mathbf{s}) \to c_{(k,gwl)}(\mathbf{s})$ and then prove it is a one-to-one mapping (i.e., no two different sources are mapped to the same target). Consider the color $c_{(k,wl)}(\mathbf{s})$ of k-tuple $\mathbf{s}$ at some iteration $t$ obtained by the k-WL algorithm (in Section 3). It is determined by the k-tuple $\mathbf{s}$ itself and its neighbors (more exactly, their colors at iteration $t-1$), which we call the stem k-tuple set $\{\mathbf{s}\} \cup \{\theta_j(\mathbf{s}, w) \mid w \in \mathcal{V}(\mathcal{G}), 1 \leq j \leq k\}$. We model the mapping from $c_{(k,wl)}(\mathbf{s})$ to its stem k-tuple set as an injective function $f$. In order to connect to k-GWL, we transform the stem k-tuple set into another set $\{\mathbf{s}\} \cup \{(\{\mathbf{s}\} \cup \{\theta_j(\mathbf{s}, w) \mid w \in \mathcal{V}(\mathcal{G})\}) \mid 1 \leq j \leq k\}$, by including $\mathbf{s}$ and $N_{(j,r)}(\mathbf{s})$ into a hyperedge. The new set can then be regarded as the k-tuple $\mathbf{s}$ and its hyperedge neighbors, which can determine the color $c_{(k,gwl)}(\mathbf{s})$ of k-tuple $\mathbf{s}$ obtained by the k-GWL algorithm. We denote by the injectve function $g$ the mapping from the new set to $c_{(k,gwl)}(\mathbf{s})$. Then the function $\phi: c_{(k,wl)}(\mathbf{s}) \to c_{(k,gwl)}(\mathbf{s})$ can be written as a composition function of $g$ and $f$, $\phi = g \circ f$.

Next, we prove $\forall c_{(k,wl)}(\mathbf{s}_1), c_{(k,wl)}(\mathbf{s}_2) \in \mathbf{C}_{(k,wl)}$, $\phi(c_{(k,wl)}(\mathbf{s}_1)) = \phi(c_{(k,wl)}(\mathbf{s}_2)) \Rightarrow c_{(k,wl)}(\mathbf{s}_1) = c_{(k,wl)}(\mathbf{s}_2)$, where $\mathbf{C}_{(k,wl)}$ denotes all possible k-tuples' final colors obtained by k-WL. Assume for contradiction that there are two final colors $c_{(k,wl)}(\mathbf{s}_1), c_{(k,wl)}(\mathbf{s}_2)$ such that $c_{(k,wl)}(\mathbf{s}_1) \neq c_{(k,wl)}(\mathbf{s}_2)$ and $\phi(c_{(k,wl)}(\mathbf{s}_1)) = \phi(c_{(k,wl)}(\mathbf{s}_2))$. It should be noted that the injective function $g$ is reversible, getting inversion function $g^{-1}$. With $g^{-1}$, the two same final colors can be transformed into the same set $\{\mathbf{s}\} \cup \{(\{\mathbf{s}\} \cup \{\theta_j(\mathbf{s}, w) \mid w \in \mathcal{V}(\mathcal{G})\}) \mid 1 \leq j \leq k\}$, which further derives the same stem k-tuple set $\{\mathbf{s}\} \cup \{\theta_j(\mathbf{s}, w) \mid w \in \mathcal{V}(\mathcal{G}), 1 \leq j \leq k\}$. Similarly, we can apply the inverse function of $f$ to get the same color of $c_{(k,wl)}(\mathbf{s}_1)$ and $c_{(k,wl)}(\mathbf{s}_2)$. Since the assumption $c_{(k,wl)}(\mathbf{s}_1) \neq c_{(k,wl)}(\mathbf{s}_2)$ leads to a contradiction, it is concluded that the injection holds.

**Proof of surjection:** Here we prove that the two algorithms yield the same number of unique k-tuple colors in every iteration. That is, there is no $c_{(k,gwl)}(\mathbf{s})$ in k-GWL that cannot be mapped from a $c_{(k,wl)}(\mathbf{s})$ in k-WL. For the first iteration $t = 0$, both two algorithms initialize k-tuple color with the *isomorphism type*. The isomorphism type of a subgraph is determined by two factors: vertex labels and the subgraph structure. Since the vertex labels processed by the two algorithms are consistent in the same graph $\mathcal{G}$, we focus primarily on the structural difference of the subgraphs. Because of the devised sub-hypergraph extraction method, it is guaranteed that the subgraph and sub-hypergraph induced by the same set of vertices in a k-tuple are the same. With the same vertex labels and subgraph structure, the two algorithms output the same k-tuple coloring.

For iteration $t > 0$, the color $c_{(k,wl)}(\mathbf{s})$ of k-tuple $\mathbf{s}$ in k-WL is determined by the stem k-tuple set $\{\mathbf{s}\} \cup \{\theta_j(\mathbf{s}, w) \mid w \in \mathcal{V}(\mathcal{G}), 1 \leq j \leq k\}$. As shown in the proof of injection, the color $c_{(k,gwl)}(\mathbf{s})$ in k-GWL is also determined by an equivalent k-tuple set $\{\mathbf{s}\} \cup \{(\{\mathbf{s}\} \cup \{\theta_j(\mathbf{s}, w) \mid w \in \mathcal{V}(\mathcal{G})\}) \mid 1 \leq j \leq k\}$. Thus, both the mapping of $c_{(k,wl)}(\mathbf{s})$ and $c_{(k,gwl)}(\mathbf{s})$ from the same k-tuple set are *injective*. Combining the injective property with that the initial colorings in iteration $t = 0$ are the same, we have in iteration $t > 0$, the number of unique colors in the colorings induced by both algorithms is also the same. Therefore, the surjection holds. Finally, since both surjection and injection hold, according to the definition of the bijection, Theorem 5.1 holds. □

For the rest proofs in this section, we use the following notations. Given two functions $f, g$, we use $f \to g$ to denote that for arbitrary input hypergraphs $H, G$, we have $f(H) = f(G) \Rightarrow g(H) = g(G)$. Then $f \to g$ implies that $f \succeq g$. When both $f \to g$ and $g \to f$ hold, there exists a bijective mapping between $f$ and $g$.

**Theorem A.2.** *(Theorem 5.2 in the main text)* $\forall k \geq 1$, *(k+1)-GOWL $\succ$ k-GOWL.*

*Proof.* k-GOWL uses the isomorphism type of k-tuple to initialize colors, as defined in Definition 4.1. We first prove that $\forall \mathbf{s} \in V^{k+1}, c_{k+1}^t(\mathbf{s}) \to c_k^t(\mathbf{s}_{:k+1})$ by enumerating $t$. Here, $c_{k+1}^t(\mathbf{s}, \mathcal{HG})$ denotes the color of the k-tuple $\mathbf{s}$ in hypergraph $\mathcal{HG}$ after applying (k+1)-GOWL for $t$ iterations.

1. For $t = 0$, the color is an isomorphism type. $\forall \mathcal{HG}_1 = (V_1, H_1, X_1), \mathcal{HG}_2 = (V_2, H_2, X_2), \forall s^1 \in V_1^{k+1}, \mathbf{s}^2 \in V_2^{k+1}$, we have

$$
\begin{aligned}
c_{k+1}^0(\mathbf{s}^1) = c_{k+1}^0(\mathbf{s}^2) \Rightarrow \\
\left(\forall i_1, i_2 \in [k+1], \mathbf{s}_{i_1}^1 = \mathbf{s}_{i_2}^1 \leftrightarrow \mathbf{s}_{i_1}^2 = \mathbf{s}_{i_2}^2\right) \wedge \left(\forall i \in [k+1], X_{1,\mathbf{s}_i^1} = X_{2,\mathbf{s}_i^2}\right) \wedge \\
\left(\forall i_1, ..., i_n \in [k+1], (\mathbf{s}_{i_1}^1, ..., \mathbf{s}_{i_n}^1) \in H^1 \leftrightarrow (\mathbf{s}_{i_1}^2, ..., \mathbf{s}_{i_n}^2) \in H^2\right) \Rightarrow \\
\left(\forall i_1, i_2 \in [k], s_{i_1}^1 = \mathbf{s}_{i_2}^1 \leftrightarrow \mathbf{s}_{i_1}^2 = \mathbf{s}_{i_2}^2\right) \wedge \left(\forall i \in [k], X_{1,\mathbf{s}_i^1} = X_{2,\mathbf{s}_i^2}\right) \wedge \\
\left(\forall i_1, ..., i_n \in [k], (\mathbf{s}_{i_1}^1, ..., \mathbf{s}_{i_n}^1) \in H^1 \leftrightarrow (\mathbf{s}_{i_1}^2, ..., \mathbf{s}_{i_n}^2) \in H^2\right) \Rightarrow \\
c_k^0(\mathbf{s}_{:k+1}^1) = c_k^0(\mathbf{s}_{:k+1}^2)
\end{aligned}
\tag{12}
$$

   Therefore, $c_{k+1}^0(\mathbf{s}, \mathcal{HG}) \to c_k^0(\mathbf{s}_{:k+1}, \mathcal{HG})$

2. $\forall t > 0$,

   In the k-GOWL algorithm, we get

$$
\begin{aligned}
c_{k+1}^{(t)}(\mathbf{e}) = \Big(\text{HASH}((c_{k+1}^{(t-1)}(\mathbf{e}), \{\!\{c_{k+1}^{(t-1)}(ns) \mid ns \in \mathcal{N}_\mathbf{s}(\mathbf{e}), \mathcal{E}_{k+1,o} = \\
\{\!\{(\{\mathbf{s}\} \cup \{\theta_j(\mathbf{s}, w) \mid w \in \mathcal{V}(\mathcal{HG}))\}) \mid 1 \leq j \leq k+1, \mathbf{s} \in \mathcal{V}(\mathcal{HG})^{k+1}\}\!\}\}\!\}))\Big) \to \\
c_k^{(t)}(\mathbf{e}') = \Big(\text{HASH}((c_k^{(t-1)}(\mathbf{e}'), \{\!\{c_k^{(t-1)}(ns) \mid ns \in \mathcal{N}_\mathbf{s}(\mathbf{e}'), \mathcal{E}_{k,o} = \\
\{\!\{(\{\mathbf{s}_{:k+1}\} \cup \{\theta_j(\mathbf{s}_{:k+1}, w) \mid w \in \mathcal{V}(\mathcal{HG}))\}) \mid 1 \leq j \leq k, \mathbf{s}_{:k+1} \in \mathcal{V}(\mathcal{HG})^k\}\!\}\}\!\}))\Big) \to \\
c_k^{(t)}(\mathbf{e}')
\end{aligned}
\tag{13}
$$

   In the above, $\mathbf{e}'$ is from $\mathcal{E}_{k,o}$ which consists of only the first k tuples of (k+1)-tuples in $\mathcal{E}_{k+1,o}$. Furthermore, we have

$$
\begin{aligned}
c_{k+1}^{(t)}(\mathbf{s}) = \Big(\text{HASH}((c_{k+1}^{(t-1)}(\mathbf{s}), \{\!\{c_{k+1}^{(t)}(ne) \mid ne \in \mathcal{N}_\mathbf{e}(\mathbf{s}), \mathcal{E}_{k+1,o} = \\
\{\!\{(\{\mathbf{s}\} \cup \{\theta_j(\mathbf{s}, w) \mid w \in \mathcal{V}(\mathcal{HG}))\}) \mid 1 \leq j \leq k+1, \mathbf{s} \in \mathcal{V}(\mathcal{HG})^{k+1}\}\!\}\}\!\}))\Big) \to \\
c_k^{(t)}(\mathbf{s}_{:k+1}) = \Big(\text{HASH}((c_k^{(t-1)}(\mathbf{s}_{:k+1}), \{\!\{c_{k+1}^{(t)}(ne) \mid ne \in \mathcal{N}_\mathbf{e}(\mathbf{s}_{:k+1}), \mathcal{E}_{k,o} = \\
\{\!\{(\{\mathbf{s}_{:k+1}\} \cup \{\theta_j(\mathbf{s}_{:k+1}, w) \mid w \in \mathcal{V}(\mathcal{HG}))\}) \mid 1 \leq j \leq k, \mathbf{s}_{:k+1} \in \mathcal{V}(\mathcal{HG})^k\}\!\}\}\!\}))\Big) \to \\
c_k^{(t)}(\mathbf{s}_{:k+1})
\end{aligned}
\tag{14}
$$

Since $c_{k+1}^{(t)}(\mathbf{s}) \to c_k^{(t)}(\mathbf{s}_{:k+1})$ holds for any iteration $t$ in k-GOWL, it follows that (k+1)-GOWL $\succeq$ k-GOWL.

It remains to prove that (k+1)-GOWL $\not\cong$ k-GOWL, i.e., there exists a pair of hypergraphs such that (k+1)-GOWL can distinguish while k-GOWL cannot. It is well-known that there exists a pair of graphs such that (k+1)-WL can distinguish while k-WL cannot (Cai et al., 1992). Since graphs are a special case of hypergraphs and k-GWL degenerates to k-WL for these inputs, we can re-use those (hyper-)graphs as the hard instances for k-GOWL which can still be distinguished by (k+1)-GOWL, completing the proof. $\square$

**Theorem A.3.** *(Theorem 5.3 in the main text)* $\forall k \geq 1$, *(k+1)-GFWL $\succ$ k-GFWL.*

*Proof.* k-GFWL uses the isomorphism type of k-tuple to initialize colors, as defined in Definition 4.1. We first prove that $\forall \mathbf{s} \in V^{k+1}, c_{k+1}^{t}(\mathbf{s}) \to c_{k}^{t}(\mathbf{s}_{:k+1})$ by enumerating $t$. Here, $c_{k+1}^{t}(\mathbf{s}, \mathcal{HG})$ denotes the color of the k-tuple $\mathbf{s}$ in hypergraph $\mathcal{HG}$ after applying (k+1)-GFWL for $t$ iterations.

1. For $t = 0$, the color is an isomorphism type. We can re-use Step 1 in the proof of Theorem 5.2 to prove $c_{k+1}^{0}(\mathbf{s}, \mathcal{HG}) \to c_{k}^{0}(\mathbf{s}_{:k+1}, \mathcal{HG})$, since k-GFWL and k-GOWL share the same initialization procedure.

2. $\forall t > 0$,

   In the k-GFWL algorithm, we get

$$c_{k+1}^{(t)}(\mathbf{e}) = \Big( \text{HASH}((c_{k+1}^{(t-1)}(\mathbf{e}), \{\!\{ c_{k+1}^{(t-1)}(ns) \mid ns \in \mathcal{N}_{\mathbf{s}}(\mathbf{e}), \mathcal{E}_{k+1,f} =$$
$$\{\!\{ (\mathbf{s}, \theta_1(\mathbf{s}, w), \theta_2(\mathbf{s}, w), \cdots, \theta_{k+1}(\mathbf{s}, w)) \mid w \in \mathcal{V}(\mathcal{HG}), \mathbf{s} \in \mathcal{V}(\mathcal{HG})^{k+1} \}\!\} \}\!\})) \Big) \to$$
$$c_{k}^{(t)}(\mathbf{e}') = \Big( \text{HASH}((c_{k}^{(t-1)}(\mathbf{e}'), \{\!\{ c_{k}^{(t-1)}(ns) \mid ns \in \mathcal{N}_{\mathbf{s}}(\mathbf{e}'), \mathcal{E}_{k,f} = \tag{15}$$
$$\{\!\{ (\mathbf{s}_{:k+1}, \theta_1(\mathbf{s}_{:k+1}, w), \theta_2(\mathbf{s}_{:k+1}, w), \cdots, \theta_k(\mathbf{s}_{:k+1}, w)) \mid w \in \mathcal{V}(\mathcal{HG}), \mathbf{s}_{:k+1} \in \mathcal{V}(\mathcal{HG})^{k} \}\!\} \}\!\})) \Big) \to$$
$$c_{k}^{(t)}(\mathbf{e}')$$

In the above, $\mathbf{e}'$ is from $\mathcal{E}_{k,f}$ which consists of only the first k tuples of (k+1)-tuples in $\mathcal{E}_{k+1,f}$. Furthermore, we have

$$c_{k+1}^{(t)}(\mathbf{s}) = \Big( \text{HASH}((c_{k+1}^{(t-1)}(\mathbf{s}), \{\!\{ c_{k+1}^{(t)}(ne) \mid ne \in \mathcal{N}_{\mathbf{e}}(\mathbf{s}), \mathcal{E}_{k+1,f} =$$
$$\{\!\{ (\mathbf{s}, \theta_1(\mathbf{s}, w), \theta_2(\mathbf{s}, w), \cdots, \theta_{k+1}(\mathbf{s}, w)) \mid w \in \mathcal{V}(\mathcal{HG}), \mathbf{s} \in \mathcal{V}(\mathcal{HG})^{k+1} \}\!\} \}\!\})) \Big) \to$$
$$c_{k}^{(t)}(\mathbf{s}_{:k+1}) = \Big( \text{HASH}((c_{k}^{(t-1)}(\mathbf{s}_{:k+1}), \{\!\{ c_{k+1}^{(t)}(ne) \mid ne \in \mathcal{N}_{\mathbf{e}}(\mathbf{s}_{:k+1}), \mathcal{E}_{k,f} = \tag{16}$$
$$\{\!\{ (\mathbf{s}_{:k+1}, \theta_1(\mathbf{s}_{:k+1}, w), \theta_2(\mathbf{s}_{:k+1}, w), \cdots, \theta_k(\mathbf{s}_{:k+1}, w)) \mid w \in \mathcal{V}(\mathcal{HG}), \mathbf{s}_{:k+1} \in \mathcal{V}(\mathcal{HG})^{k} \}\!\} \}\!\})) \Big) \to$$
$$c_{k}^{(t)}(\mathbf{s}_{:k+1})$$

Since $c_{k+1}^{(t)}(\mathbf{s}) \to c_{k}^{(t)}(\mathbf{s}_{:k+1})$ holds for any iteration $t$ in k-GFWL, it follows that (k+1)-GFWL $\succeq$ k-GFWL.

It remains to prove that (k+1)-GFWL $\not\cong$ k-GFWL. We can apply the similar proof idea in Theorem 5.2 to get hard instances for k-GFWL which can be distinguished by (k+1)-GFWL, completing the proof. $\square$

We first present the following lemma that will be used in the proof of Theorem 5.4 and the lemma will be proved shortly.

**Lemma A.4.** *Let $k \geq 2$. For all hypergraphs $\mathcal{HG}, \mathcal{HG}'$, all $\mathbf{v}, \mathbf{v}' \in \mathcal{V}(\mathcal{HG})^{k+1}$ and all iterations $t \geq 0$, $c_{o,k+1}^{(t)}(\mathbf{v}) = c_{o,k+1}^{(t)}(\mathbf{v}')$ is equivalent to that $ISO_{k+1}(\mathbf{v}) = ISO_{k+1}(\mathbf{v}')$ and $c_{f,k}^{(t)}(\theta_i(\mathbf{v},)) = c_{f,k}^{(t)}(\theta_i(\mathbf{v}',))$ for all $1 \leq i \leq k+1$.*

**Theorem A.5.** *(Theorem 5.4 in the main text)* $\forall k \geq 2$, *(k+1)-GOWL $\cong$ k-GFWL.*

*Proof.* Let $c_{f,k}^{(t)}$ and $c_{o,k}^{(t)}$ be the coloring of k-GFWL and k-GOWL in the $t^{th}$ iteration, respectively. We prove that for all hypergraphs $\mathcal{HG}, \mathcal{HG}'$ and all iterations $t \geq 0$, (1) if $\mathcal{HG}, \mathcal{HG}'$ are distinguished by $c_{f,k}^{(t)}$, they are distinguished by $c_{o,k+1}^{(t)}$; and (2) if $\mathcal{HG}, \mathcal{HG}'$ are distinguished by $c_{o,k+1}^{(t)}$, they are distinguished by $c_{f,k}^{(t+1)}$.

To prove (1), suppose $\mathcal{HG}, \mathcal{HG}'$ are not distinguished by $c_{o,k+1}^{(t)}$. Then there is a bijection $f \colon \mathcal{V}(\mathcal{HG})^{k+1} \to \mathcal{V}(\mathcal{HG}')^{k+1}$ such that $c_{o,k+1}^{(t)}(\mathbf{v}) = c_{o,k+1}^{(t)}(f(\mathbf{v}))$ for all $\mathbf{v} \in \mathcal{V}(\mathcal{HG})^{k+1}$.

For $\mathbf{u} = (u_1, u_2, \cdots, u_k)$, let $\mathbf{u}_+ = (u_1, u_2, \cdots, u_k, u_k)$ and $\mathbf{u}'_+ = (u'_1, u'_2, \cdots, u'_k, u'_{k+1}) = f(\mathbf{u}_+)$. According to $c^{(t)}_{o,k+1}(\mathbf{u}_+) = c^{(t)}_{o,k+1}(\mathbf{u}'_+)$ and Lemma A.4, we get $ISO_{k+1}(\mathbf{u}_+) = ISO_{k+1}(\mathbf{u}'_+)$ and thus $u'_{k+1} = u'_k$. Let a mapping $g(\mathbf{u}) = (u'_1, u'_2, \cdots, u'_k) = \mathbf{u}'$. Then $g$ is a bijection from $\mathcal{V}(\mathcal{HG})^k$ to $\mathcal{V}(\mathcal{HG}')^k$. Since $\mathbf{u} = \theta_{k+1}(\mathbf{u}_+,)$ and $\mathbf{u}' = \theta_{k+1}(\mathbf{u}'_+,)$, we have $c^{(t)}_{f,k}(\mathbf{u}_+) = c^{(t)}_{f,k}(\mathbf{u}'_+)$ by applying Lemma A.4 in the position $k+1$. Then $g$ is a bijection that preserves the coloring $c^{(t)}_{f,k}$ and thus $c^{(t)}_{f,k}$ cannot distinguish $\mathcal{HG}, \mathcal{HG}'$.

To prove (2), suppose $\mathcal{HG}, \mathcal{HG}'$ are not distinguished by $c^{(t+1)}_{f,k}$. Then there is a bijection $g: \mathcal{V}(\mathcal{HG})^k \to \mathcal{V}(\mathcal{HG}')^k$ such that $c^{(t+1)}_{f,k}(\mathbf{u}) = c^{(t+1)}_{f,k}(g(\mathbf{u}))$ for all $\mathbf{u} \in \mathcal{V}(\mathcal{HG})^k$. Let $\mathbf{u}' = g(\mathbf{u})$. By the definition of $c^{(t+1)}_{f,k}$ in k-GFWL, we have $c^{(t)}_{f,k}(\mathbf{u}) = c^{(t)}_{f,k}(\mathbf{u})$ and there is a bijection $h : \mathcal{V}(\mathcal{HG}) \to \mathcal{V}(\mathcal{HG}')$ such that for all $w \in \mathcal{V}(\mathcal{HG})$, we have $ISO_{k+1}(\mathbf{u}w) = ISO_{k+1}(\mathbf{u}'h(w))$ and $c^{(t)}_{f,k}(\theta_i(\mathbf{u}, w)) = c^{(t)}_{f,k}(\theta_i(\mathbf{u}', h(w)))$ for all $1 \le i \le k$. This implies $c^{(t)}_{f,k}(\theta_i(\mathbf{u}w,)) = c^{(t)}_{f,k}(\theta_i(\mathbf{u}'h(w),))$ for all $1 \le i \le k$. It is clearly true that $c^{(t)}_{f,k}(\theta_{k+1}(\mathbf{u}w,)) = c^{(t)}_{f,k}(\theta_{k+1}(\mathbf{u}'h(w),))$. Then by Lemma A.4, we get that $c^{(t)}_{o,k+1}(\mathbf{u}w) = c^{(t)}_{o,k+1}(\mathbf{u}'h(w))$. We can then define a bijection $f: \mathcal{V}(\mathcal{HG})^{k+1} \to \mathcal{V}(\mathcal{HG}')^{k+1}$ such that $f(\mathbf{u}w) = g(\mathbf{u})h(w)$ for all $\mathbf{u} \in \mathcal{V}(\mathcal{HG})^k$ and all $w \in \mathcal{V}(\mathcal{HG})$. $f$ preserves the coloring $c^{(t)}_{o,k+1}$ and thus $c^{(t)}_{o,k+1}$ cannot distinguish $\mathcal{HG}, \mathcal{HG}'$. $\qquad\square$

We require $k \ge 2$ to ensure that the isomorphism type function ISO() is activated in k-GFWL, since it has certain distinguishing capability as shown in Figure 3(a). We now prove Lemma A.4.

*Proof.* (Proof of Lemma A.4) We prove the equivalence by induction on iteration $t$. The base case $t = 0$ is trivial. For the inductive step $t \to t+1$, to prove the forward direction, we suppose that $c^{(t+1)}_{o,k+1}(\mathbf{v}) = c^{(t+1)}_{o,k+1}(\mathbf{v}')$. By the definition of $c^{(t+1)}_{o,k+1}$, we get $ISO_{k+1}(\mathbf{v}) = ISO_{k+1}(\mathbf{v}')$. For $1 \le i \le k+1$, it remains to prove $c^{(t+1)}_{f,k}(\theta_i(\mathbf{v},)) = c^{(t+1)}_{f,k}(\theta_i(\mathbf{v}',))$. By the definition of $c^{(t+1)}_{o,k+1}$, we have $\{\!\{c^{(t)}_{o,k+1}(\theta_i(\mathbf{v}, w)) \mid w \in \mathcal{V}(\mathcal{HG})\}\!\} = \{\!\{c^{(t)}_{o,k+1}(\theta_i(\mathbf{v}', w')) \mid w' \in \mathcal{V}(\mathcal{HG}')\}\!\}$. Then there is a bijection $h : \mathcal{V}(\mathcal{HG}) \to \mathcal{V}(\mathcal{HG}')$ such that $c^{(t)}_{o,k+1}(\theta_i(\mathbf{v}, w)) = c^{(t)}_{o,k+1}(\theta_i(\mathbf{v}', h(w)))$ for all $w \in \mathcal{V}(\mathcal{HG})$.

According to the inductive hypothesis, we have that $ISO_{k+1}(\theta_i(\mathbf{v}, w)) = ISO_{k+1}(\theta_i(\mathbf{v}', h(w)))$ and $c^{(t)}_{f,k}(\theta_j(\theta_i(\mathbf{v}, w),)) = c^{(t)}_{f,k}(\theta_j(\theta_i(\mathbf{v}', h(w)),))$ for all $1 \le j \le k+1$. When $j = i$, this implies $c^{(t)}_{f,k}(\theta_i(\mathbf{v},)) = c^{(t)}_{f,k}(\theta_i(\mathbf{v}',))$. And for all $1 \le j \le k$, we have $c^{(t)}_{f,k}(\theta_j(\theta_i(\mathbf{v},), w)) = c^{(t)}_{f,k}(\theta_j(\theta_i(\mathbf{v}',), h(w)))$. By the definition of k-GFWL, it follows that $c^{(t+1)}_{f,k}(\theta_i(\mathbf{v},)) = c^{(t+1)}_{f,k}(\theta_i(\mathbf{v}',))$.

For the backward direction, we suppose that $ISO_{k+1}(\mathbf{v}) = ISO_{k+1}(\mathbf{v}')$ and $c^{(t+1)}_{f,k}(\theta_i(\mathbf{v},)) = c^{(t+1)}_{f,k}(\theta_i(\mathbf{v}',))$ for all $1 \le i \le k+1$. Since $c^{(t+1)}_{f,k} \to c^{(t)}_{f,k}$, by the inductive hypothesis, it follows that $c^{(t)}_{o,k+1}(\mathbf{v}) = c^{(t)}_{o,k+1}(\mathbf{v}')$. By definition of $c^{(t+1)}_{o,k+1}$, to prove $c^{(t+1)}_{o,k+1}(\mathbf{v}) = c^{(t+1)}_{o,k+1}(\mathbf{v}')$, we need to prove that $\{\!\{c^{(t)}_{o,k+1}(\theta_i(\mathbf{v}, w)) \mid w \in \mathcal{V}(\mathcal{HG})\}\!\} = \{\!\{c^{(t)}_{o,k+1}(\theta_i(\mathbf{v}', w')) \mid w' \in \mathcal{V}(\mathcal{HG}')\}\!\}$ for all $1 \le i \le k+1$.

Consider only an $i$ for $1 \le i \le k+1$. Because $c^{(t+1)}_{f,k}(\theta_i(\mathbf{v},)) = c^{(t+1)}_{f,k}(\theta_i(\mathbf{v}',))$, there is a bijection $h : \mathcal{V}(\mathcal{HG}) \to \mathcal{V}(\mathcal{HG}')$ such that for all $w \in \mathcal{V}(\mathcal{HG})$, we have $ISO_{k+1}(\theta_i(\mathbf{v},)w) = ISO_{k+1}(\theta_i(\mathbf{v}',)h(w))$ and $c^{(t)}_{f,k}(\theta_j(\theta_i(\mathbf{v},), w)) = c^{(t)}_{f,k}(\theta_j(\theta_i(\mathbf{v}',), h(w)))$ for all $1 \le j \le k$. This implies that $ISO_{k+1}(\theta_i(\mathbf{v}, w)) = ISO_{k+1}(\theta_i(\mathbf{v}', h(w)))$ and $c^{(t)}_{f,k}(\theta_j(\theta_i(\mathbf{v}, w),)) = c^{(t)}_{f,k}(\theta_j(\theta_i(\mathbf{v}', h(w)),))$ for all $1 \le j \le k+1$. Then by the inductive hypothesis, we get that $c^{(t)}_{o,k+1}(\theta_i(\mathbf{v}, w)) = c^{(t)}_{o,k+1}(\theta_i(\mathbf{v}', h(w)))$. Since $h$ is a bijection, it holds that $\{\!\{c^{(t)}_{o,k+1}(\theta_i(\mathbf{v}, w)) \mid w \in \mathcal{V}(\mathcal{HG})\}\!\} = \{\!\{c^{(t)}_{o,k+1}(\theta_i(\mathbf{v}', w')) \mid w' \in \mathcal{V}(\mathcal{HG}')\}\!\}$, completing the proof. $\qquad\square$

**Theorem A.6.** *(Theorem 6.1 in the main text) Given a hypergraph $\mathcal{HG} = \{\mathcal{V}, \mathcal{E}\}$ and let $k \ge 2$. Then for all iterations $t \ge 0$, for initial features $x^{(0)}_\mathbf{s}$ consistent with the initial colorings $c^{(0)}_k$ of k-GWL and for all weights $W^{(t)}$,*

$$c^{(t)}_k \sqsubseteq x^{(t)}_\mathbf{s} \tag{17}$$

*Proof.* We prove for an arbitrary iteration $t$ and k-tuples $\mathbf{s}, \mathbf{s}' \in \mathcal{V}(\mathcal{HG})^k$, that $c^{(t)}_k(\mathbf{s}) = c^{(t)}_k(\mathbf{s}')$ implies $x^{(t)}_\mathbf{s} = x^{(t)}_{\mathbf{s}'}$. In fact, we can also prove for hyperedges $\mathbf{e}$ and $\mathbf{e}'$ in $\mathcal{HG}_k$, $c^{(t)}_k(\mathbf{e}) = c^{(t)}_k(\mathbf{e}')$ implies $x^{(t)}_\mathbf{e} = x^{(t)}_{\mathbf{e}'}$. In iteration $t = 0$, we

have $c_k^{(0)}(\mathbf{s}) = c_k^{(0)}(\mathbf{s}') \Rightarrow x_{\mathbf{s}}^{(0)} = x_{\mathbf{s}'}^{(0)}$ since the initial features $x_{\mathbf{s}}^{(0)}$ are chosen consistent with the initial colorings $c_k^{(0)}$ of k-GWL. Similarly, $c_k^{(0)}(\mathbf{e}) = c_k^{(0)}(\mathbf{e}') \Rightarrow x_{\mathbf{e}}^{(0)} = x_{\mathbf{e}'}^{(0)}$ because the initial hyperedge coloring and feature vectors are set to be uniform.

Let k-tuple $\mathbf{s}, \mathbf{s}' \in \mathcal{V}(\mathcal{HG})^k$ and $t > 0$ such that $c_k^{(t)}(\mathbf{s}) = c_k^{(t)}(\mathbf{s}')$. Suppose for the induction that $c_k^{(t-1)}(\mathbf{s}) = c_k^{(t-1)}(\mathbf{s}') \Rightarrow x_{\mathbf{s}}^{(t-1)} = x_{\mathbf{s}'}^{(t-1)}$ and $c_k^{(t-1)}(\mathbf{e}) = c_k^{(t-1)}(\mathbf{e}') \Rightarrow x_{\mathbf{e}}^{(t-1)} = x_{\mathbf{e}'}^{(t-1)}$ hold. As $c_k^{(t)}(\mathbf{s}) = c_k^{(t)}(\mathbf{s}')$, we know from the k-GWL refinement step that the old colors $c_k^{(t-1)}(\mathbf{s}) = c_k^{(t-1)}(\mathbf{s}')$ of $\mathbf{s}$ and $\mathbf{s}'$ as well as hyperedge color multisets $\{\!\!\{c_k^{(t)}(\mathbf{e}) \,|\, \mathbf{e} \in \mathcal{N}_{\mathbf{e}}(\mathbf{s})\}\!\!\}$ and $\{\!\!\{c_k^{(t)}(\mathbf{e}) \,|\, \mathbf{e} \in \mathcal{N}_{\mathbf{e}}(\mathbf{s}')\}\!\!\}$ of the hyperedge neighbors of $\mathbf{s}$ and $\mathbf{s}'$ are identical. The latter, according to the refinement step, implies that the multisets $P_{\mathbf{s}} = \{\!\!\{c_k^{(t-1)}(\theta_j(\mathbf{s}, w)) \,|\, 1 \le j \le k, w \in V(\mathcal{G})\}\!\!\}$ and $P_{\mathbf{s}}' = \{\!\!\{c_k^{(t-1)}(\theta_j(\mathbf{s}', w)) \,|\, 1 \le j \le k, w \in V(\mathcal{G})\}\!\!\}$ are identical (together with that the colors of the hyperedges at iteration $t-1$, $\{\!\!\{c_k^{(t-1)}(\mathbf{e}) \,|\, \mathbf{e} \in \mathcal{N}_{\mathbf{e}}(\mathbf{s})\}\!\!\}$ and $\{\!\!\{c_k^{(t-1)}(\mathbf{e}) \,|\, \mathbf{e} \in \mathcal{N}_{\mathbf{e}}(\mathbf{s}')\}\!\!\}$ are the same).

Let $Q_{\mathbf{s}} = \{\!\!\{x_{\theta_j(\mathbf{s}, w)}^{(t-1)} \,|\, 1 \le j \le k, w \in V(\mathcal{HG})\}\!\!\}$ and $Q_{\mathbf{s}'} = \{\!\!\{x_{\theta_j(\mathbf{s}', w)}^{(t-1)} \,|\, 1 \le j \le k, w \in V(\mathcal{HG})\}\!\!\}$ be the multisets of feature vectors (corresponding to $P_{\mathbf{s}}$ and $P_{\mathbf{s}}'$ with colors replaced by feature vectors). By the inductive hypothesis, we know that $x_{\mathbf{s}}^{(t-1)} = x_{\mathbf{s}'}^{(t-1)}$, $Q_{\mathbf{s}} = Q_{\mathbf{s}'}$, and $\{\!\!\{x_{\mathbf{e}}^{(t-1)} \,|\, \mathbf{e} \in \mathcal{N}_{\mathbf{e}}(\mathbf{s})\}\!\!\} = \{\!\!\{x_{\mathbf{e}}^{(t-1)} \,|\, \mathbf{e} \in \mathcal{N}_{\mathbf{e}}(\mathbf{s}')\}\!\!\}$. The latter two imply that $\{\!\!\{x_{\mathbf{e}}^{(t)} \,|\, \mathbf{e} \in \mathcal{N}_{\mathbf{e}}(\mathbf{s})\}\!\!\} = \{\!\!\{x_{\mathbf{e}}^{(t)} \,|\, \mathbf{e} \in \mathcal{N}_{\mathbf{e}}(\mathbf{s}')\}\!\!\}$. Then, regardless of the choice of the aggregation and updating functions in k-HNNs, we get $x_{\mathbf{s}}^{(t)} = x_{\mathbf{s}'}^{(t)}$ since the input of the functions is the same. This proves $c_k^{(t)}(\mathbf{s}) = c_k^{(t)}(\mathbf{s}') \Rightarrow x_{\mathbf{s}}^{(t)} = x_{\mathbf{s}'}^{(t)}$ and thus $c_k^{(t)} \sqsubseteq x_{\mathbf{s}}^{(t)}$. $\qquad \square$

**Theorem A.7.** *(Theorem 6.2 in the main text) Let $\mathcal{A} : \mathcal{HG} \to \mathcal{R}^d$ be a k-HNN based on k-tuple neighborhood. With a sufficient number of layers, for any pair of hypergraphs that k-GWL determines as non-isomorphic, $\mathcal{A}$ also determines them as non-isomorphic if (1) the aggregation and update functions in the base HNN of $\mathcal{A}$ are injective, and (2) $\mathcal{A}$'s hypergraph embedding readout function is injective.*

*Proof.* Let $\mathcal{HG}, \mathcal{HG}'$ be any pair of hypergraphs that k-GWL determines as non-isomorphic at iteration $K$. Since $\mathcal{A}$'s hypergraph embedding readout function is injective, it suffices to prove that $\mathcal{A}$'s two-stage neighborhood aggregation process, with sufficient iterations, embeds $\mathcal{HG}$ and $\mathcal{HG}'$ into different multisets of k-tuple embeddings. Suppose $\mathcal{A}$ updates k-tuple representations as follows:

$$x_{\mathbf{e}}^{(t)} = g'\big(x_{\mathbf{e}}^{(t-1)}, f'(\{\!\!\{x_{\mathbf{s}}^{(t-1)} \,|\, \mathbf{s} \in \mathcal{N}_{\mathbf{s}}(\mathbf{e})\}\!\!\})\big)$$
$$x_{\mathbf{s}}^{(t)} = g\big(x_{\mathbf{s}}^{(t-1)}, f(\{\!\!\{x_{\mathbf{e}}^{(t)} \,|\, \mathbf{e} \in \mathcal{N}_{\mathbf{e}}(\mathbf{s})\}\!\!\})\big),$$

where $g, g'$ and $f, f'$ functions are injective.

The k-GWL applies an injective hash function $h$ to update the k-GWL k-tuple labels:

$$c_k^{(t)}(\mathbf{e}) = h'\big((c_k^{(t-1)}(\mathbf{e}), \{\!\!\{c_k^{(t-1)}(\mathbf{s}) \,|\, \mathbf{s} \in \mathcal{N}_{\mathbf{s}}(\mathbf{e})\}\!\!\})\big)$$
$$c_k^{(t)}(\mathbf{s}) = h\big((c_k^{(t-1)}(\mathbf{s}), \{\!\!\{c_k^{(t)}(\mathbf{e}) \,|\, \mathbf{e} \in \mathcal{N}_{\mathbf{e}}(\mathbf{s})\}\!\!\})\big).$$

We will prove, by induction, that for any iteration $t$, there always exists an injective function $\sigma$ such that

$$x_{\mathbf{s}}^{(t)} = \sigma(c_k^{(t)}(\mathbf{s})). \tag{18}$$

In addition, we prove that for any iteration $t$, there always exists an injective function $\sigma'$ such that

$$x_{\mathbf{e}}^{(t)} = \sigma'(c_k^{(t)}(\mathbf{e})). \tag{19}$$

For the base case $t = 0$, Eq. (18) trivially holds since the initial features are set based on the isomorphism type using Eq. (9) which is an injective function. Eq. (19) also holds since the initial hyperedge features in k-HNN are the same as the hyperedge labels in k-GWL. Suppose that Eq. (18) and (19) hold for iteration $t - 1$, we show they also hold for $t$. With substitutions, the updating of hyperedge representations becomes

$$x_{\mathbf{e}}^{(t)} = g'\big(\sigma'(c_k^{(t-1)}(\mathbf{e})), f'(\{\!\{\sigma(c_k^{(t-1)}(\mathbf{s})) \mid \mathbf{s} \in \mathcal{N}_{\mathbf{s}}(\mathbf{e})\}\!\})\big).$$

Since the composition of injective functions is injective, there exists some injective function $\psi'$ such that

$$x_{\mathbf{e}}^{(t)} = \psi'\big(c_k^{(t-1)}(\mathbf{e}), \{\!\{c_k^{(t-1)}(\mathbf{s}) \mid \mathbf{s} \in \mathcal{N}_{\mathbf{s}}(\mathbf{e})\}\!\}\big).$$

We further get that

$$\begin{aligned}
x_{\mathbf{e}}^{(t)} &= \psi'\big(c_k^{(t-1)}(\mathbf{e}), \{\!\{c_k^{(t-1)}(\mathbf{s}) \mid \mathbf{s} \in \mathcal{N}_{\mathbf{s}}(\mathbf{e})\}\!\}\big) \\
&= \psi' \cdot (h')^{-1} \cdot h'\big(c_k^{(t-1)}(\mathbf{e}), \{\!\{c_k^{(t-1)}(\mathbf{s}) \mid \mathbf{s} \in \mathcal{N}_{\mathbf{s}}(\mathbf{e})\}\!\}\big) \\
&= \psi' \cdot (h')^{-1} \cdot c_k^{(t)}(\mathbf{e}).
\end{aligned}$$

Then there exists an injective function $\sigma' = \psi' \cdot (h')^{-1}$ such that $x_{\mathbf{e}}^{(t)} = \sigma'(c_k^{(t)}(\mathbf{e}))$. We can then apply similar trick in the updating of k-tuple representations:

$$x_{\mathbf{s}}^{(t)} = g\big(\sigma(c_k^{(t-1)}(\mathbf{s})), f(\{\!\{\sigma'(c_k^{(t)}(\mathbf{e})) \mid \mathbf{e} \in \mathcal{N}_{\mathbf{e}}(\mathbf{s})\}\!\})\big).$$

Because the composition of injective functions is injective, there exists some injective function $\psi$ such that

$$\begin{aligned}
x_{\mathbf{s}}^{(t)} &= \psi\big(c_k^{(t-1)}(\mathbf{s}), \{\!\{c_k^{(t)}(\mathbf{e}) \mid \mathbf{e} \in \mathcal{N}_{\mathbf{e}}(\mathbf{s})\}\!\}\big) \\
&= \psi \cdot h^{-1} \cdot h\big(c_k^{(t-1)}(\mathbf{s}), \{\!\{c_k^{(t)}(\mathbf{e}) \mid \mathbf{e} \in \mathcal{N}_{\mathbf{e}}(\mathbf{s})\}\!\}\big) \\
&= \psi \cdot h^{-1} \cdot c_k^{(t)}(\mathbf{s}).
\end{aligned}$$

Hence, there exists an injective function $\sigma = \psi \cdot h^{-1}$ such that $x_{\mathbf{s}}^{(t)} = \sigma(c_k^{(t)}(\mathbf{s}))$. Therefore, at iteration $K$, when k-GWL determines $\mathcal{HG}, \mathcal{HG}'$ as non-isomorphic with different k-tuple labels $\{\!\{c_k^{(K)}(\mathbf{s})\}\!\}$, k-HNN must also have different features $\{\!\{x_{\mathbf{s}}^{(K)}\}\!\} = \{\!\{\sigma(c_k^{(K)}(\mathbf{s}))\}\!\}$ and decide them as non-isomorphic. $\qquad\square$

## B. The high-dimensional Weisfeiler-Lehman test of hypergraph isomorphism algorithm

---

**Algorithm 1** The high-dimensional Weisfeiler-Lehman test of hypergraph isomorphism algorithm

---

**Input:** Hypergraphs $\mathcal{HG}$ and $\mathcal{HG}'$, int k: dimensionailty, ISO($\cdot$): isomorphism intial function, $\mathcal{HG}_{sub}(\mathbf{s})$: sub-hypergraph induced by k-tuple $\mathbf{s}$ from $\mathcal{HG}$, k-tuples hypergraph $\mathcal{HG}_k$, int $h$: maximum number of iterations

// 1. Initialize k-tuple s' colors in $\mathcal{V}(\mathcal{HG})^k$.

$c_k^{(0)}(\mathbf{s}) \leftarrow \text{ISO}(\mathcal{HG}_{sub}(\mathbf{s})), \mathbf{s} \in \mathcal{V}(\mathcal{HG})^k$

$c_k^{(0)}(\mathbf{s}') \leftarrow \text{ISO}(\mathcal{HG}_{sub}(\mathbf{s}')), \mathbf{s} \in \mathcal{V}(\mathcal{HG}')^k$

// 2. Initialize hyperedge s' color in hypergraph $\mathcal{HG}_k$ for k-tuples.

$c_k^{(0)}(\mathbf{e}) \leftarrow 0, \mathbf{e} \in \mathcal{E}(\mathcal{HG}_k)$

$c_k^{(0)}(\mathbf{e}') \leftarrow 0, \mathbf{e}' \in \mathcal{E}(\mathcal{HG}'_k)$

$t \leftarrow 0$

**while** $t < h$ and $\{\!\!\{ c_k^{(t)}(\mathbf{s}) \mid \mathbf{s} \in \mathcal{V}(\mathcal{HG})^k \}\!\!\} = \{\!\!\{ c_k^{(t)}(\mathbf{s}') \mid \mathbf{s}' \in \mathcal{V}(\mathcal{HG}')^k \}\!\!\}$ **do**

  $t \leftarrow t + 1$

  // 3. Gathering hyperedges' k-tuple neighbors to color hyperedges.

  **for** $\mathbf{e}/\mathbf{e}' \in \mathcal{E}(\mathcal{HG}_k)/\mathcal{E}(\mathcal{HG}'_k)$ **do**

    $c_k^{(t)}(\mathbf{e}) \leftarrow \text{HASH}\big((c_k^{(t-1)}(\mathbf{e}), \{\!\!\{ c_k^{(t-1)}(\mathbf{u}) \mid \mathbf{u} \in \mathcal{N}_\mathbf{s}(\mathbf{e}) \}\!\!\})\big)$

    $c_k^{(t)}(\mathbf{e}') \leftarrow \text{HASH}\big((c_k^{(t-1)}(\mathbf{e}'), \{\!\!\{ c_k^{(t-1)}(\mathbf{u}) \mid \mathbf{u} \in \mathcal{N}_\mathbf{s}(\mathbf{e}') \}\!\!\})\big)$

  **end for**

  // 4. Gathering k-tuples' hyperedge neighbors to color k-tuples.

  **for** $\mathbf{s}/\mathbf{s}' \in \mathcal{V}(\mathcal{HG})^k/V(\mathcal{HG}')^k$ **do**

    $c_k^{(t)}(\mathbf{s}) \leftarrow \text{HASH}\big((c_k^{(t-1)}(\mathbf{s}), \{\!\!\{ c_k^{(t)}(\mathbf{u}) \mid \mathbf{u} \in \mathcal{N}_\mathbf{e}(\mathbf{s}) \}\!\!\})\big)$

    $c_k^{(t)}(\mathbf{s}') \leftarrow \text{HASH}\big((c_k^{(t-1)}(\mathbf{s}'), \{\!\!\{ c_k^{(t)}(\mathbf{u}) \mid \mathbf{u} \in \mathcal{N}_\mathbf{e}(\mathbf{s}') \}\!\!\})\big)$

  **end for**

**end while**

Comparing $\{\!\!\{ c_k^{(t)}(\mathbf{s}) \mid \mathbf{s} \in \mathcal{V}(\mathcal{HG})^k \}\!\!\}$ and $\{\!\!\{ c_k^{(t)}(\mathbf{s}') \mid \mathbf{s}' \in \mathcal{V}(\mathcal{HG}')^k \}\!\!\}$

**Ensure:** Whether $\mathcal{HG}$ and $\mathcal{HG}'$ are isomorphic or not.

---

## C. Sub-Hypergraph Extractions

We reveal that induced sub-hypergraphs, while appearing straightforward, require careful consideration. Recently there is a growing interest in learning a good representation for sub-hypergraphs in hypergraphs, e.g., SHINE (Luo, 2022) in bioinformatic applications. Here we propose a sub-hypergraph extraction trick and discuss the differences between sub-hypergraphs and traditional subgraphs.

First, let us review the method for extracting subgraphs based on given vertices in a simple graph. Given graph $\mathcal{G}(X, A)$ with vertex features $X \in \mathbb{R}^{|\mathcal{V}| \times F}$ and adjacency matrix $A \in \{0, 1\}^{|\mathcal{V}| \times |\mathcal{V}|}$. Given a set of vertices $\mathbf{s} = \{v_1, ..., v_i\}$, our purpose is to extract subgraph $\mathcal{G}_{sub}(X_{sub}, A_{sub})$ based on vertices $\mathbf{s}$ from $\mathcal{G}$. It is easy to obtain $X_{sub} \in \mathbb{R}^{|\mathbf{s}| \times F}$ by selecting the vertex features from $\mathbf{s}$. The edge relationships among the vertex set $\mathbf{s}$ can be obtained by retaining only the relevant rows and columns of the adjacency matrix $A$, resulting in $A_{sub} \in \{0, 1\}^{|\mathbf{s}| \times |\mathbf{s}|}$.

Then, in the hypergraph $\mathcal{HG}(X, H)$ with vertex features $X \in \mathbb{R}^{|\mathcal{V}| \times F}$ and incidence matrix $H \in \{0, 1\}^{|V| \times |E|}$, the features of the sub-hypergraph's vertices $X_{sub} \in \mathbb{R}^{|\mathbf{s}| \times F}$ can be obtained by selecting the features corresponding to the vertex set $\mathbf{s}$. The main issue lies in how we derive the sub-hypergraph's incidence matrix $H_{sub}$. One could simply retain the relevant rows in the incidence matrix $H$ and delete columns/hyperedges that contain no vertex in $\mathbf{s}$, referred to as $H'$. But we propose two (equivalent) rules to prune $H'$:

1. If a hyperedge in the sub-hypergraph $H'$ contains at least two vertices from the vertex set in its original hyperedge, then the hyperedge is retained. (In other words, columns with at least two non-zero entries are kept in $H'$.)

2. If a hyperedge only contains a single vertex, called a singleton hyperedge, then the hyperedge is deleted from $H'$. (In other words, columns with only one non-zero entry are pruned from $H'$.)

Therefore, we obtain $H_{sub} \in \{0, 1\}^{|\mathbf{s}| \times |U|}$ by retaining the relevant rows in the incidence matrix $H$ and delete columns/hyperedges that contain no vertex or only one vertex in $\mathbf{s}$, where $|U|$ is the number of hyperedges that contain at least two vertices from the set $\mathbf{s}$.

The first rule is well-motivated in practice and easy to explain. For instance, in citation datasets, hyperedges represent papers and co-authors are the vertices within the hyperedges. When we remove some co-authors from a paper/hyperedge, the co-author relationship among the remaining authors is still maintained.

The second rule essentially ensures that the sub-hypergraph and the subgraph induced by the same set of vertices in an equivalent hypergraph/graph are the same. See the illustrative example in Figure 1. In fact, when there is a single vertex in a hyperedge, the hyperedge information can be either deleted or transformed into an additional vertex feature. We turn to the former, which facilitates the theoretical proof of the relation between k-GWL and k-WL. Additionally, from the experimental results shown in Table 2, we find that deleting singleton hyperedges significantly enhances the hypergraph classification performance compared to retaining them. We believe that our finding on sub-hypergraph extractions can be of independent interest.

# D. Illustrative Examples for Hypergraph Isomorphism Test

For the examples in this section, we only consider k-sets for clarity. The omitted ordering (e.g., $(v_1, v_2)$ vs. $(v_2, v_1)$) and vertex repetitions (e.g., $(v_1, v_1)$) do not affect the correctness. Symmetric 2-tuples have the same features since they have the same induced sub-hypergraph. 2-tuples with vertex repetitions have an induced sub-hypergraph as the vertex itself since singleton hyperedges are removed.

## D.1. An Example Distinguishable During the Initialization of 2-GWL

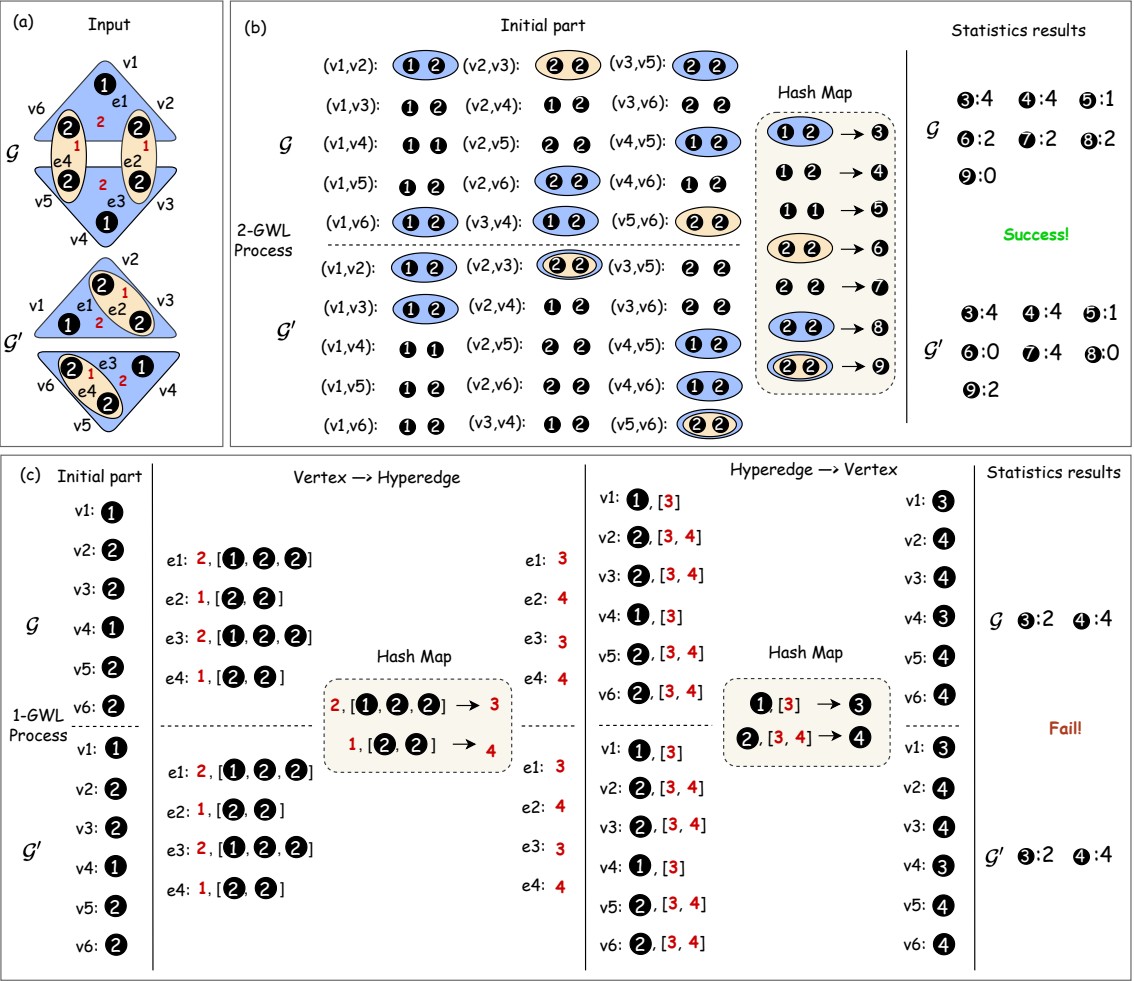

*Figure 5.* An illustrative example where 2-GWL can distinguish and 1-GWL cannot distinguish. Node labels are in circles and edge labels are of red color.

In part (a), two non-isomorphic hypergraphs, $G$ and $G'$, are presented. Part (b) illustrates the initialization process of 2-GWL, where 2-GWL successfully identifies distinct subgraphs in $G$ and $G'$. For example, the subgraph type generated by $(v_5, v_6)$ in $G'$ is unique and does not match any subgraph type in $G$, allowing us to conclude that they are non-isomorphic.

In contrast, part (c) shows that 1-GWL is unable to distinguish between the two hypergraphs. After each round of 1-GWL, the color mapping remains the same. Thus, 1-GWL fails to distinguish between these two non-isomorphic hypergraphs. Therefore, the expressive power of 2-GWL is indeed stronger than that of 1-GWL.

### D.2. A Harder Example Not Distinguishable During the Initialization of 2-GWL

We have presented an example where two non-isomorphic hypergraphs can be distinguished during the initialization phase of 2-GWL in Figure 5. To further validate the expressive power of the $k$-GWL algorithm, we provide a more challenging example of non-isomorphic hypergraphs that cannot be distinguished during the initialization phase of 2-GWL. Below, we present the details of applying 1-GWL, 2-GOWL, and 2-GFWL in the hypergraphs.

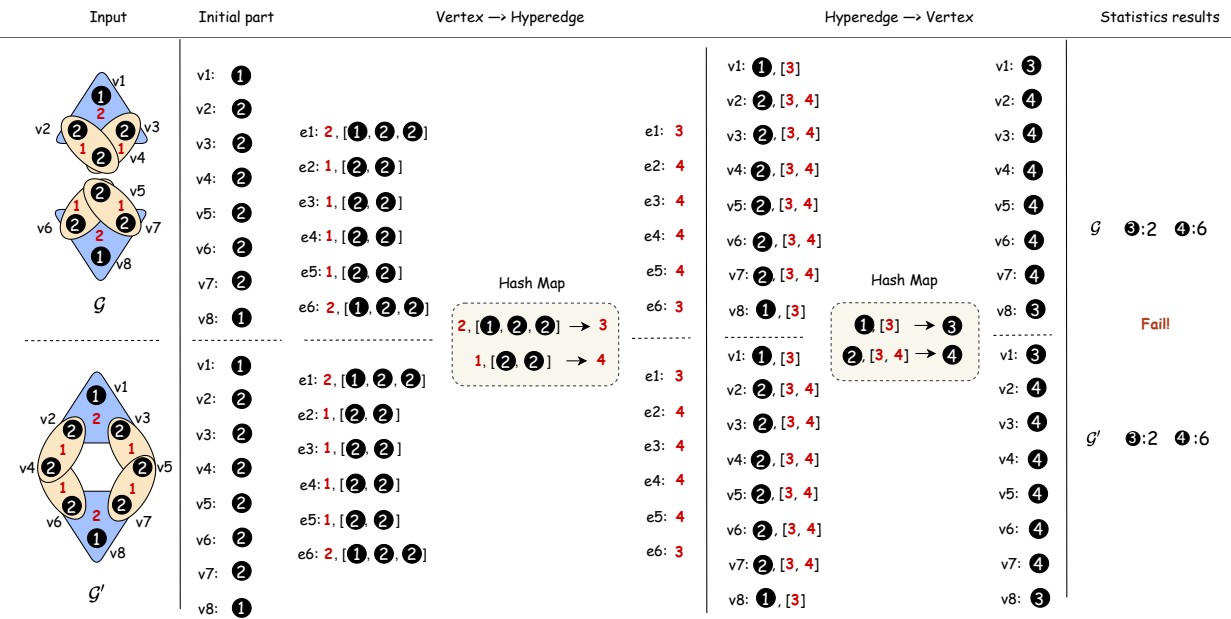

*Figure 6.* Illustration of 1-GWL failing to distinguish two particular non-isomorphic hypergraphs. Node labels are in circles and edge labels are of red color.

In Figure 6, we show that 1-GWL is unable to distinguish the challenging example of non-isomorphic hypergraphs. Similarly to part (c) in Figure 5, after each round of 1-GWL, the types and quantities of colors generated remain consistent with those from the previous round. Therefore, 1-GWL fails to distinguish these two non-isomorphic hypergraphs.

In the following, we show that 2-GFWL successfully distinguishes the non-isomorphic hypergraphs while 2-GOWL cannot distinguish. We start with the details of the initialization part of 2-GWL (shared by 2-GFWL and 2-GOWL), as shown in Figure 7.

We observe that cardinalities of the 2-tuple colors in the two hypergraphs keep the same, although the colors for $(v_3, v_4)$, $(v_3, v_5)$, $(v_4, v_6)$ and $(v_5, v_6)$ are different. This indicates that neither 2-GFWL nor 2-GOWL can distinguish these two non-isomorphic hypergraphs during the initialization phase.

In the next step, our 2-GOWL and 2-GFWL algorithms construct hyperedges for each initialized 2-tuple to connect it with its corresponding 2-tuple hypergraph neighbors. For brevity, we only consider the 2-tuples $(v_3, v_4)$ and $(v_3, v_5)$ and construct their 2-tuple hypergraph neighbors. The 2-tuple hypergraphs and their isomorphism test processes are shown in Figure 8 and Figure 9, respectively.

In Figure 8, we use the hyperedge construction formula of k-GOWL mentioned in Section 4.1.2 as follows:

$$\mathcal{E}_{k,o} = \{\!\{\big(\{\mathbf{s}\} \cup \{\theta_j(\mathbf{s}, w) \,|\, w \in \mathcal{V}(\mathcal{HG})\}\big) \,|\, 1 \leq j \leq k, \, \mathbf{s} \in \mathcal{V}(\mathcal{HG})^k\}\!\}$$

It can be deemed as replacing a specific vertex in the k-tuple with all vertices from the original vertex set to form a k-tuple neighbor, and then all these k-tuple neighbors are connected to form a hyperedge. Specifically, for the 2-tuple $(v_3, v_4)$, we observe that by replacing $v_3$ in $(v_3, v_4)$ with $v_1, v_2, v_5, v_6, v_7$, and $v_8$, we obtain six 2-tuple neighbors. These six neighbors

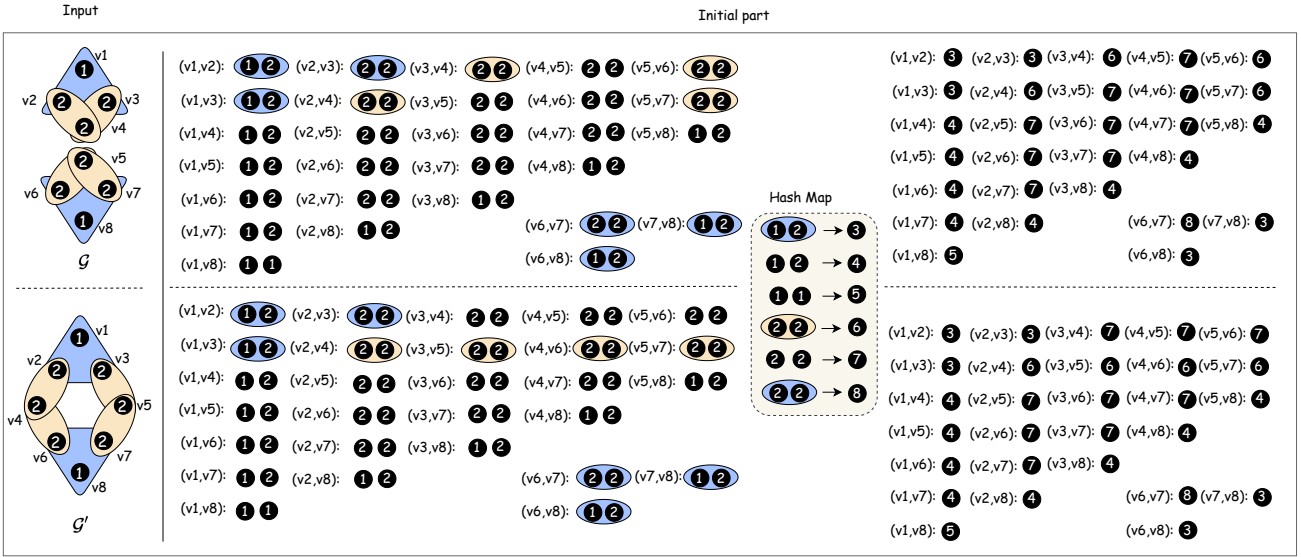

*Figure 7.* The initialization phase of the 2-GWL cannot distinguish the non-isomorphic hypergraphs as well.

are connected to $(v_3, v_4)$ by the hyperedge $e_1$, forming parts of the 2-tuple hypergraph. The construction process for $(v_3, v_5)$ follows similarly.

Next, we perform the 1-GWL color relabeling on the constructed 2-tuple hypergraph. As a result, we obtain that in $G$, 2-tuple $(v_3, v_4)$ is assigned color '9' and $(v_3, v_5)$ is assigned color '10'. In $G'$, 2-tuple $(v_3, v_4)$ is assigned color '10' and $(v_3, v_5)$ is assigned color '9'. The cardinalities of colors for these two 2-tuples remain consistent with the input. Therefore, 2-GOWL is unable to distinguish these two non-isomorphic hypergraphs.

In Figure 9, k-GFWL constructs the set of hyperedges as follows:

$$\mathcal{E}_{k,f} = \{\!\!\{ \big( \mathbf{s}, \theta_1(\mathbf{s}, w), \theta_2(\mathbf{s}, w), \cdots, \theta_k(\mathbf{s}, w) \big) \mid w \in \mathcal{V}(\mathcal{HG}), \, \mathbf{s} \in \mathcal{V}(\mathcal{HG})^k \}\!\!\}$$

It can be considered as replacing every vertex in the k-tuple with a specific vertex from the original vertex set to form a k-tuple neighbor, and then all these k-tuple neighbors are connected to form a hyperedge. Specifically, for the 2-tuple $(v_3, v_4)$, we replace both $v_3$ and $v_4$ with $v_1$, resulting in the 2-tuple neighbors $(v_1, v_3)$ and $(v_1, v_4)$. These two neighbors, along with $(v_3, v_4)$, are connected by the hyperedge $e_1$. The construction process for $(v_3, v_5)$ is similar.

Finally, we perform the 1-GWL coloring on the constructed 2-tuple hypergraph. The results are that in $G$, $(v_3, v_4)$ is assigned color '9' and $(v_3, v_5)$ is assigned color '10'. In $G'$, $(v_3, v_4)$ is assigned color '11' and $(v_3, v_5)$ is assigned color '12'. Since the 2-tuple colors in the two non-isomorphic hypergraphs differ, the 2-GFWL algorithm successfully distinguishes between the two hypergraphs.

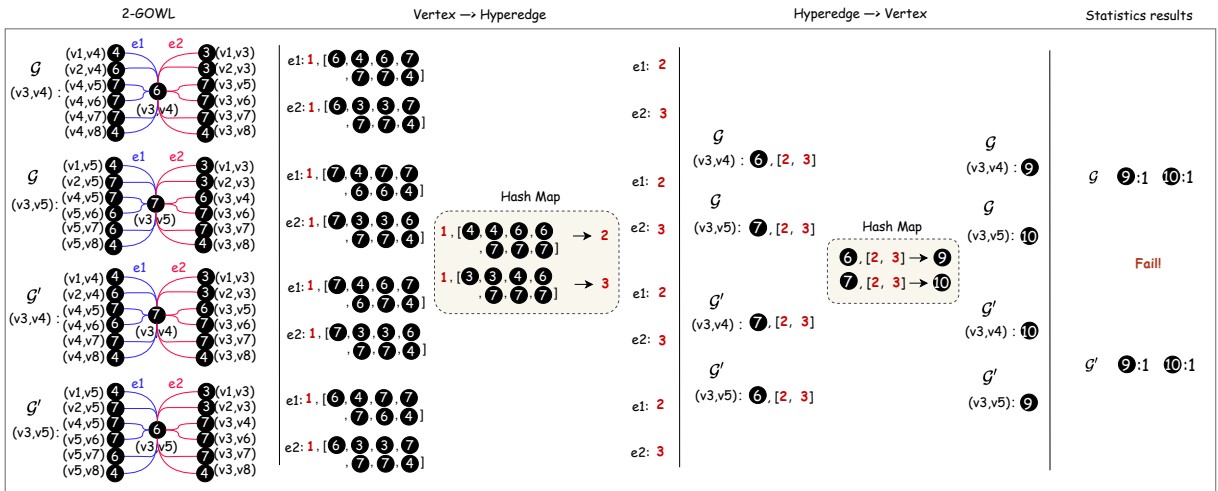

*Figure 8.* Illustration of 2-GOWL failing to distinguish non-isomorphic hypergraphs.

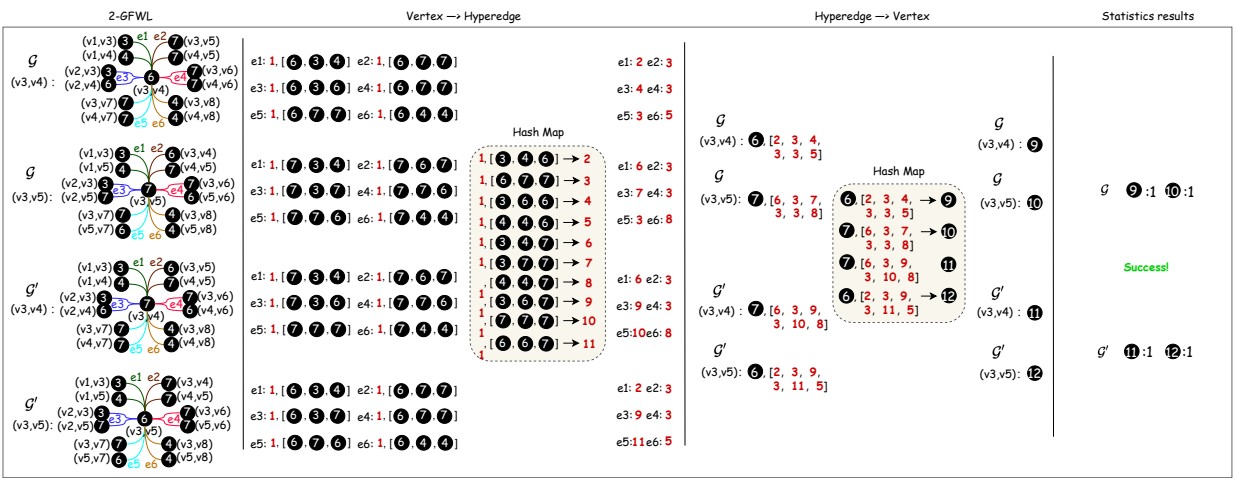

*Figure 9.* Illustration of 2-GFWL successfully distinguishing non-isomorphic hypergraphs.

# E. Experiment Details

## E.1. Experimental Setups

**Datasets:** In the experiments, we adopt three types of real-world hypergraph classification datasets proposed in HIC [14], which are IMDB, Steam-Player, and Twitter-Friend. In the IMDB datasets, two correlations—co-director and co-writer—are used for high-order dataset construction. In the dataset name, "Dir" and "Wri" indicate hypergraphs constructed from the co-director and co-writer relationships, respectively. The staff (director/writer) of each movie forms a hypergraph. "Form" categorizes movies by type (e.g., animation, drama), while "Genre" classifies them by genre (e.g., adventure, crime). The Steam-Player dataset comprises games played by Steam users, represented as hypergraphs, where vertices represent games played, and hyperedges connect games with shared tags. The goal of the dataset is to identify each user's preference: single-player or multi-player games. The Twitter-Friend dataset is a social media dataset where each hypergraph represents a user's friends. Hyperedges connect users who are friends. The label indicates whether the user posted about "National Dog Day" or "Respect Tyler Joseph". To assess the efficacy of capturing distinct correlation structures, all datasets exclude the original vertex features. Full dataset statistics are in Table 5, where $|v|$ and $|e|$ represent the number of vertices and the number of hyperedges in a hypergraph, respectively. $\overline{d}_{\mathbf{v}}$ and $\overline{d}_{\mathbf{e}}$ are the average degrees of vertices and hyperedges, respectively. In addition, we provide the information of k-tuple hypergraphs generated by the 2-GWL algorithm, which are presented in Table 6 and Table 7, respectively.

*Table 5.* Statistics of Input Hypergraph Datasets

| DATASET | # HYPERGRAPHS | # CLASSES | AVG.$|v|$ | AVG.$|e|$ | MAX.$|v|$ | MAX.$|e|$ | $\overline{d}_{\mathbf{e}}$ | $\overline{d}_{\mathbf{v}}$ |
|---|---|---|---|---|---|---|---|---|
| IMDB-DIR-FORM | 1869 | 3 | 15.7 | 39.2 | 264 | 450 | 3.7 | 6.3 |
| IMDB-DIR-GENRE | 3393 | 3 | 17.3 | 36.4 | 264 | 450 | 3.8 | 5.9 |
| IMDB-WRI-FORM | 374 | 4 | 10.1 | 3.7 | 180 | 23 | 5.0 | 1.5 |
| IMDB-WRI-GENRE | 1172 | 6 | 12.8 | 4.4 | 273 | 75 | 5.2 | 1.5 |
| STEAM-PLAYER | 2048 | 2 | 13.8 | 46.4 | 76 | 140 | 4.5 | 14.9 |
| TWITTER-FRIEND | 1310 | 2 | 21.6 | 84.3 | 166 | 603 | 4.3 | 16.4 |

*Table 6.* Statistics of k-Tuple Hypergraphs by 2-GOWL

| DATASET | AVG.$|v|$ | AVG.$|e|$ | $\overline{d}_{\mathbf{e}}$ | $\overline{d}_{\mathbf{v}}$ |
|---|---|---|---|---|
| IMDB-DIR-FORM | 223.4 | 368.1 | 9.2 | 12.6 |
| IMDB-DIR-GENRE | 301.8 | 477.2 | 9.6 | 12.7 |
| IMDB-WRI-FORM | 117.6 | 44.5 | 6.4 | 3.6 |
| IMDB-WRI-GENRE | 266.8 | 94.7 | 8.3 | 3.9 |
| STEAM-PLAYER | 96.8 | 21.7 | 12.6 | 2.9 |
| TWITTER-FRIEND | 345.2 | 611.5 | 13.7 | 21.2 |

*Table 7.* Statistics of k-Tuple Hypergraphs by 2-GFWL

| DATASET | AVG.$|v|$ | AVG.$|e|$ | $\overline{d}_{\mathbf{e}}$ | $\overline{d}_{\mathbf{v}}$ |
|---|---|---|---|---|
| IMDB-DIR-FORM | 223.4 | 6773.3 | 2.5 | 14.2 |
| IMDB-DIR-GENRE | 301.8 | 9068.1 | 2.5 | 14.8 |
| IMDB-WRI-FORM | 117.6 | 3435.15 | 2.6 | 7.7 |
| IMDB-WRI-GENRE | 266.8 | 10009.7 | 2.6 | 10.2 |
| STEAM-PLAYER | 96.8 | 511.0 | 3.0 | 12.0 |
| TWITTER-FRIEND | 345.2 | 7684.9 | 2.5 | 22.1 |

**Compared Methods:** In our k-HNN, the number of neighbors for each k-set grows exponentially with k, which is constrained by the GPU memory capacity. Therefore, we mainly focus on $k = 2$ (except for 3-HNNs with vertex sampling). Due to the two different neighbor aggregation approaches, we obtain two variants of our models: k-FHNN and k-OHNN. Since our models are end-to-end trained HNNs with node features and hypergraph structure as input and final hypergraph features as output, we choose several representative hypergraph deep learning methods as baselines. These include: HyperGCN (Yadati et al., 2019), HNHN (Dong et al., 2020), UniGNNII (Huang & Yang, 2021), AllSetTransformer (Chien

et al., 2022), and ED-HNN (Wang et al., 2023). Furthermore, we adapt the 1-GWL kernel kernel method (Feng et al., 2024) to an end-to-end model, resulting in HIC as another baseline. To further emphasize the contribution of structural information to the learning process, we also include the 2-layer MLP of width 128 as a baseline model. Note that some of the above methods were originally proposed for node classification tasks. Therefore, we add the same readout module in their final step, where the embeddings of all nodes are summed to obtain the final hypergraph embedding used for classification.

**Other Details:** To evaluate the model's performance, we report the average classification accuracy over 5-fold cross-validation along with standard deviation. This metric intuitively measures the model's ability to distinguish different types of hypergraphs. We use the Adam optimizer with a learning rate of 0.001 and terminate the training if there is no improvement in the validation performance after 50 epochs. All experiments are conducted on a server with an Intel E5-2650 CPU (2.20GHz), 256GB RAM, and an NVIDIA 1080Ti GPU (11GB), except the experiment for 3-HNNs on a GPU cluster consisting of 10 Sugon X640 G30 GPU servers, including 1 management node and 9 GPU compute nodes. Each node is equipped with dual Intel Xeon 6248R CPU, 512GB RAM, and 8 Nvidia Tesla A100 GPU(40GB).

### E.2. Main Result

The experimental results for three types of real-world hypergraph datasets are presented in Table 1. Based on those results, we have four observations. First, our proposed two HNN models outperform other compared methods across all datasets. Notably, on the Steam-Player dataset, 2-FHNN outperforms the runner-up model, AllSetTransformer, by 6% in accuracy. This advantage stems from our model, guided by k-GWL, being able to better capture the high-order structures within hypergraph data. Second, we have run experiments on k-HNNs for $k = 3$ based on the vertex sampling strategy with sample size 10 in a GPU cluster with 40GB memory. Although not the full 3-HNNs without the sampling, 3-HNNs with the sampling have comparable performance to 2-HNNs across the datasets, while significantly outperforming in the IMDB-Wri-Genre dataset. Third, in the IMDB-Wri-Form dataset, the structure-free MLP model achieves a relatively high ranking, trailing our model by only 3% in accuracy. This indicates that leveraging high-order structural information in this dataset still has a large space for improvement. Fourth, although 2-FHNN is theoretically more advantageous than 2-OHNN, experimental results reveal mixed performance across datasets. For instance, 2-FHNN performs better on datasets such as IMDB-Dir-Form, IMDB-Wri-Form, and Steam-Player, while 2-OHNN outperforms on IMDB-Dir-Genre, IMDB-Wri-Genre, and Twitter-Friend datasets. Therefore, in practical applications, it is essential to select the model that best suits the specific scenario.

### E.3. Impact of Sub-hypergraph Extraction Methods

We here conduct comparative experiments for different sub-hypergraph extraction methods to show the proposed induced sub-hypergraph method not only supports the unification of graphs and hypergraphs, but also offers empirical advantages. We append "-S" to our model when keeping all hyperedges, including singleton hyperedges, in extracting sub-hypergraphs. Note that this only changes the initial features of k-sets, i.e., isomorphism types, due to a possibly different induced sub-hypergraph, while the k-set hypergraph structure remains unchanged. The experimental results are presented in Table 2. We observe that when k-sets use sub-hypergraphs containing a large number of singleton hyperedges as the isomorphism type features, the model performs poorly across all datasets. However, when we remove singleton hyperedges from the sub-hypergraphs, the model achieves an average performance improvement of 5.4% across all datasets, with a maximum improvement of 14%. This highlights the indispensable role of our induced sub-hypergraph module and its significant impact on the empirical performance of k-HNN.

### E.4. Influence of the Vertex Sampling Strategy

In k-GWL, the time complexity inevitably grows exponentially with the increase in $k$. To reduce the complexity of k-tuple hypergraph construction, we propose a vertex sampling strategy. Specifically, we sort the vertex set of each hypergraph dataset by degree and select the top $m$ vertices with the highest degrees. These sampled vertices are then used to construct the k-tuple hypergraph and train the models, while the remaining vertices are ignored.

Considering the distribution of vertex numbers across the datasets, we set $m$ to 5, 10, 15, and 20, resulting in four experimental groups. Additionally, we included a fully sampled experiment (using all vertices) as a baseline for comparison. We conduct the same sampling experiments on both the 2-FHNN and 2-OHNN models. The experimental results for different models are shown in Table 3 and Table 6, respectively.

*Table 3.* Results on Sample Vertex Numbers (2-FHNN)

|  | 5 | 10 | 15 | 20 | ALL |
|---|---|---|---|---|---|
| IMDB-DIR-FORM | 66.45±0.98 | 65.97±2.54 | 66.51±1.39 | 66.67±2.83 | **68.11±2.46** |
| IMDB-DIR-GENRE | 76.24±0.98 | 77.49±1.23 | 77.40±1.44 | 77.04±2.67 | **78.52±1.07** |
| IMDB-WRI-FORM | 52.22±3.99 | 52.78±4.39 | 48.61±5.20 | 51.11±5.23 | **55.36±3.30** |
| IMDB-WRI-GENRE | **47.93±3.26** | 43.10±2.33 | 38.36±1.98 | 37.76±1.58 | 45.44±1.24 |
| STEAM-PLAYER | 61.81±2.20 | 63.43±1.71 | 65.88±1.91 | 65.98±1.51 | **67.53±1.23** |
| TWITTER-FRIEND | 62.15±1.30 | 61.08±1.23 | **62.85±2.14** | 62.31±1.67 | 62.75±2.18 |

*Table 6.* Results on Sample Vertex Numbers (2-OHNN)

|  | 5 | 10 | 15 | 20 | ALL |
|---|---|---|---|---|---|
| IMDB-DIR-FORM | 66.45±1.31 | 66.72±1.50 | 67.10±1.47 | 67.15±1.27 | **67.25±2.35** |
| IMDB-DIR-GENRE | 76.39±1.85 | 75.68±1.45 | 77.49±1.43 | 75.12±1.03 | **79.75±1.14** |
| IMDB-WRI-FORM | 49.44±6.37 | 49.17±2.58 | 53.06±5.08 | 51.94±3.58 | **55.35±3.74** |
| IMDB-WRI-GENRE | 47.50±3.17 | 42.93±2.54 | 36.90±3.50 | 38.19±1.80 | **50.08±1.89** |
| STEAM-PLAYER | 60.88±1.25 | 62.94±1.01 | 65.05±0.69 | 64.75±1.77 | **65.97±1.37** |
| TWITTER-FRIEND | 61.54±1.95 | 61.92±2.81 | 62.08±1.16 | 62.85±2.59 | **64.12±1.30** |

Based on those results, we have three observations. First, for both the 2-FHNN and 2-OHNN models, the performance improves as the number of sampled vertices $m$ increases, reaching its peak when all vertices are sampled. This trend is particularly pronounced in the Steam-Player dataset, where the performance difference between the case with 5 sampled vertices and the full sampling scenario is as high as 6%. Second, in some datasets, such as IMDB-Wri-Form and IMDB-Wri-Genre, the model performance initially decreases and then increases as the number of sampled vertices increases. This may be because, in these datasets, the key structural information is concentrated in a few high-degree vertices. Adding low-degree vertices, which lack meaningful information, can interfere with learning the graph structure. Third, when the number of sampled vertices $m$ is low, 2-FHNN generally performs better than 2-OHNN. For example, in the cases where $m = 5$ and $m = 10$, 2-FHNN outperforms 2-OHNN in 9 out of the 12 experimental datasets. This suggests that 2-FHNN is more promising in the setting of aggressive pruning for high efficiency.

### E.5. Runtime Results

To investigate the time complexity of the k-HNN model, we record the average time required to run one epoch across different datasets and vertex sampling sizes. The runtime results for 2-FHNN and 2-OHNN are presented in Table 4 and Table 7, respectively. Additionally, we also recorded the average time required to run one epoch across different hypergraph learning models. The runtime results are shown in Table 8.

*Table 4.* Time per Epoch (seconds) for Different Sampling Numbers (2-FHNN)

|  | 5 | 10 | 15 | 20 | ALL |
|---|---|---|---|---|---|
| IMDB-DIR-FORM | 7.58 | 8.46 | 8.46 | 8.58 | 10.27 |
| IMDB-DIR-GENRE | 13.81 | 14.59 | 15.34 | 15.54 | 20.71 |
| IMDB-WRI-FORM | 1.49 | 1.57 | 1.59 | 1.59 | 1.72 |
| IMDB-WRI-GENRE | 4.67 | 4.86 | 4.83 | 5.06 | 5.89 |
| STEAM-PLAYER | 10.75 | 11.10 | 11.21 | 10.87 | 11.16 |
| TWITTER-FRIEND | 6.82 | 7.13 | 7.63 | 7.34 | 7.77 |

From the experimental results in Table 4 and Table 7, we make the following three observations. First, even with the same number of sampled vertices, the runtime of the same model varies significantly across different datasets. For instance, in the 2-FHNN model experiments, when sampling five vertices, the average runtime per epoch time is only 1.49 s on the IMDB-Wri-Genre dataset, while it rises to 13.81 s on the IMDB-Dir-Genre dataset, a nearly 9-fold difference. This discrepancy can likely be attributed to the considerable difference in the average degree of hyperedges in the original hypergraph topology. For example, the average hyperedge degree in the IMDB-Wri-Genre dataset is only 3.7, compared to 36.4 in the IMDB-Dir-Genre dataset, which is almost 10 times higher. As a result, even if two hypergraphs have the same number of nodes, differences in their topological structures can lead to substantial differences in model runtime.

*Table 7.* Time per Epoch (s) for Different Sampling Numbers (2-OHNN)

|  | 5 | 10 | 15 | 20 | ALL |
|---|---|---|---|---|---|
| IMDB-DIR-FORM | 7.50 | 8.30 | 8.51 | 8.50 | 8.98 |
| IMDB-DIR-GENRE | 13.65 | 14.46 | 14.90 | 15.41 | 17.41 |
| IMDB-WRI-FORM | 1.50 | 1.53 | 1.56 | 1.54 | 1.57 |
| IMDB-WRI-GENRE | 4.59 | 5.09 | 5.12 | 5.25 | 6.96 |
| STEAM-PLAYER | 9.09 | 11.34 | 11.41 | 11.32 | 11.55 |
| TWITTER-FRIEND | 6.71 | 7.10 | 7.56 | 7.43 | 7.71 |

*Table 8.* Time per Epoch (s) for Different HNN Models

|  | IMDB-DIR-FORM | IMDB-DIR-GENRE | IMDB-WRI-FORM | IMDB-WRI-GENRE | STEAM-PLAYER | TWITTER-FRIEND |
|---|---|---|---|---|---|---|
| HNHN | 7.4 | 10.78 | 0.3 | 0.87 | 2.94 | 1.37 |
| HYPERGCN | 39.54 | 131.23 | 3.8 | 13.58 | 98.49 | 100.73 |
| HIC | 5.68 | 10.27 | 0.99 | 3.16 | 5.58 | 4.48 |
| UNIGCNII | 5.52 | 9.92 | 0.98 | 3.18 | 5.4 | 3.19 |
| ALLSETTRANS | 5.8 | 10.24 | 1.01 | 3.28 | 5.69 | 4.16 |
| ED-HNN | 5.66 | 10.19 | 0.99 | 3.29 | 5.63 | 4.64 |
| 2-OHNN(OURS) | 8.98 | 17.41 | 1.57 | 6.96 | 11.55 | 7.71 |
| 2-FHNN(OURS) | 10.27 | 20.71 | 1.72 | 5.89 | 11.16 | 7.77 |

Second, as the number of sampled vertices increases, the time required for the model to complete one epoch also increases. However, the rate of increase differs across datasets. We speculate that this is related to the average degree of the hyperedges in the datasets. For instance, in the IMDB-Dir-Genre dataset, where the hyperedge degree is higher, the time to complete one epoch increases by nearly 50% when going from sampling 5 vertices to full sampling. In contrast, in the IMDB-Wri-Genre dataset, where the degree of hyperedge is lower, the time required to complete one epoch only increases by about 15%.

Third, when the number of vertices inevitably increases, using 2-OHNN will have a lower runtime compared to 2-FHNN, even though their theoretical time complexities for constructing k-tuples are the same. For example, in the IMDB-Dir-Genre dataset, under the full sampling of vertices, 2-OHNN requires 14% less time than 2-FHNN.

Based on the experimental results in Table 8, we also make three observations. First, the training time per epoch for HyperGCN is significantly higher than all the other models. For example on the Twitter-Friend dataset, it is 14 times longer than the second-most time-consuming model. This is because HyperGCN is a spectral HNN that involves expensive Laplacian computations for every hypergraph in a dataset. Note that we did not include a variant of HyperGCN (Yadati et al., 2019) that could run faster at the expense of worse embeddings.

Second, apart from HyperGCN, our 2-OHNN and 2-FHNN methods have the highest runtime. This is due to the computation of the k-set embeddings. However, the runtimes of our models do not exhibit explosive growth; they remain within an acceptable range, at most twice of the fast model UniGCNII.

Finally, combining with the runtime results of the vertex sampling experiment in Table 4 and Table 7, when the number of sampled vertices is 5, the runtimes of 2-FHNN and 2-OHNN become much closer to those of the main models. Especially in the IMDB-DIR-GENRE dataset, with 5 vertices sampled, the time ratio between our 2-FHNN and 2-OHNN models and the fastest model, UniGCNII, shrinks to within a factor of 1.4.

### E.6. Effect of Adding Node Labels

To validate whether the inclusion of additional node labels can enhance the expressive power of the model, we conduct the following experiments using the HIC model as the baseline (whose expressive power is upper bounded by 1-GWL). We select two nodes in the graph to include additional feature values of 1 and 2, while the remaining nodes receive an additional feature value of 0. To ensure that each' pair of nodes is select at least once, we generate $P(n, 2)$ label graphs, where $P(n, 2)$ represents all possible combinations of selecting 2 nodes from $n$ nodes. We aggregate the final embeddings of all label graphs process by the HIC model to obtain the final graph embedding, which is then used for graph classification tasks. The variants obtain by the HIC model using this method are referred to as (1,2)-HIC. The experimental results of HIC and (1,2)-HIC are shown in Table 9.

*Table 9.* Comparison on the Inclusion of Additional Node Labels

| MODEL | IMDB-WRI-FORM | STEAM-PLAYER |
|---|---|---|
| HIC | 49.74±5.09 | 58.35±1.01 |
| (1,2)-HIC | 53.61±3.36 | 61.15±3.08 |

The experimental results indicate that adding additional node labels can indeed improve the model's accuracy. For example, on the IMDB-Wri-Form and Steam-Player datasets, the model's performance improves by approximately 3.5% on average, further demonstrating that this method enhances the model's expressive power.

### E.7. The Number of Layers and The Number of Parameters in the Tested Models

*Table 10.* The Number of Parameters in the Models with 1 to 3 Layers

| MODEL | 1 LAYER | 2 LAYERS | 3 LAYERS |
|---|---|---|---|
| MLP | NA | 9,219 | 20,102 |
| HYPERGCN | NA | 59,395 | 87,942 |
| HNHN | NA | 125,443 | 192,131 |
| UNIGCNII | NA | 41,731 | 58,115 |
| ALLSETTRANSFORMER | 135,171 | NA | NA |
| ED-HNN | 109,443 | NA | NA |
| HIC | NA | 25,987 | 59,907 |
| K-HNNS | 117,379 | 216,963 | 316,547 |

*Table 11.* The Best-Performing Number of Layers in the Models

| MODEL | IMDB-DIR-FORM | IMDB-DIR-GENRE | IMDB-WRI-FORM | IMDB-WRI-GENRE | STEAM-PLAYER | TWITTER-FRIEND |
|---|---|---|---|---|---|---|
| MLP | 3 | 3 | 3 | 2 | 3 | 2 |
| HYPERGCN | 2 | 1 | 3 | 2 | 3 | 2 |
| HNHN | 3 | 2 | 2 | 2 | 2 | 3 |
| UNIGCNII | 3 | 3 | 3 | 2 | 3 | 2 |
| ALLSETTRANSFORMER | 1 | 1 | 1 | 1 | 1 | 1 |
| ED-HNN | 1 | 1 | 1 | 1 | 1 | 1 |
| HIC | 3 | 2 | 2 | 3 | 3 | 3 |
| 2-OHNN | 1 | 3 | 2 | 1 | 2 | 2 |
| 2-FHNN | 2 | 2 | 1 | 1 | 1 | 2 |

Here we report the number of parameters for all tested models towards a transparent comparison. We run different number of layers from 1 to 3 for each of the models and report the highest performance achieved in Table 1. The best-performing number of layers in the tested models and their corresponding number of parameters can be found in Tables 11 and 10, respectively. For MLPs used in these models, we uniformly set the number of hidden layers as 2 and their dimension as 128. In Table 10, k-HNNs have the same number of parameters for different values of k since they share the same neural network architecture. It can be observed that our k-HNN models, AllSetTransformer, and HNHN use more parameters than others. However, our proposed models and AllSetTransformer clearly outperform other baselines as shown in Table 1. For both graph and hypergraph deep learning, it is still an open problem on how to address the trade-off between expressive power and model complexity. In addition, we report the average time per epoch for all the models in Table 8 of Appendix E.5. The run-times of our methods are mostly smaller than double those of compared methods, while the vertex sampling approach can effectively reduce the run-time to be closer to that of other methods.

