# OpenReview forum: "Improved Expressivity of Hypergraph Neural Networks through High-Dimensional Generalized Weisfeiler-Leman Algorithms"
_ICML.cc/2025/Conference — ICML 2025 poster_

### Official Review · Reviewer_NmFi · 2025-02-20

**Overall Recommendation:** 2

**Summary:**

The paper presents a higher-order version of the WL test for hypergraphs. It is a conservative extension of the well-known higher-order WL for graphs, in the sense that when run over graphs, both tests have the same expressive power. A hypergraph-GNN architecture is then designed in terms of this higher-order test: It is shown that the expressive power of such hyperedge-GNNs is bounded by that of the higher-order WL test, and that it is always possible to emulate the expressive power of the test over graphs with n vertices by one of these hyperedge-GNN architectures. Experiments analyzing the suitability of the approach on real-world datasets.

**Claims And Evidence:**

All theoretical results are supported by carefully written proofs. My only concern refers to the proof that k-folklore and (k+1)-oblivious WL coincide in expressive power (see my comments in the Questions for Authors).

**Essential References Not Discussed:**

I think that the related literature is well-covered in the paper.

**Experimental Designs Or Analyses:**

I have not checked this since I do not have enough competence to do it.

**Methods And Evaluation Criteria:**

Yes, the experimental data has been chosen appropriately.

**Other Comments Or Suggestions:**

I have no further comments.

**Other Strengths And Weaknesses:**

As mentioned before, I feel lukewarm about this paper. While it is a solid piece of work and may interest some researchers in the community, it lacks truly innovative contributions. Perhaps its main strength lies in the clear presentation of the extension of k-WL from graphs to hypergraphs, but I have doubts about whether this alone is sufficient to justify acceptance at ICML.

**Questions For Authors:**

Q1. About the removal of singleton hyperedges: Does it mean that you allow no vertex colors in hypergraphs? Do you allow for repetition of vertices in hyperedges? Cannot you then encode a singleton hyperedge of the form (v) with (v,v)?

Q2. Regarding your result that k-folklore and (k+1)-oblivious WL are equally expressive: I understand that this is a proper generalization of the result obtained by Grohe & Otto for graphs. But your proof seems considerably simpler. In fact, Grohe & Otto requires heavy machinery based on linear algebra and linear programming to obtain this result. Please compare your result against them and explain how you managed to obtain the results without extending their machinery.

**Relation To Broader Scientific Literature:**

The paper extends a well-established line of research on graphs and hypergraphs, exploring the expressive power of GNNs in relation to isomorphism tests. The techniques employed are standard within this literature. In this sense, while the paper is solid and well-rounded, it is somewhat incremental and not particularly innovative. For instance, all separability results obtained in the paper are shown by a direct translation into well-known (and sophisticated) results for graphs. This raises some doubts about its suitability for ICML. In my view, it falls slightly below the standard one would expect from a strong ICML paper.

**Theoretical Claims:**

I have not read all proofs in the appendix, but I know in depth this class of results and I am basically convinced about their correctness (save for the equivalence of k-folklore and (k+1)-oblivious WL, see below).

---

> ### Author Rebuttal · Authors · 2025-03-30
>
> >As mentioned before, I feel lukewarm about this paper. While it is a solid piece of work and may interest some researchers in the community, it lacks truly innovative contributions. Perhaps its main strength lies in the clear presentation of the extension of k-WL from graphs to hypergraphs, but I have doubts about whether this alone is sufficient to justify acceptance at ICML.
>
> k-WL cannot be directly applied to hypergraphs. The two main challenges in generalizing k-WL to hypergraphs are: (1) how to initialize the features of k-tuples through sub-hypergraph extractions while ensuring degeneration to k-WL, and (2) how to construct k-tuple hypergraphs from the original hypergraphs so that 1-GWL can be applied. Our key findings include: (a) establishing a Generalized WL hierarchy for hypergraphs with increasing expressivity, with the notable difference between 1-GFWL vs 2-GOWL, unlike its graph counterpart. (b) The hierarchy allows us to design hypergraph neural networks with the desired expressivity, improves upon most existing hypergraph neural networks whose expressive power is upper bounded by 1-GWL.
>
> >Q1. About the removal of singleton hyperedges: Does it mean that you allow no vertex colors in hypergraphs? Do you allow for repetition of vertices in hyperedges? Cannot you then encode a singleton hyperedge of the form (v) with (v,v)?
>
> Thank you for the comments. We allow vertex colors/features in hypergraphs, such as the example of vertex colors 1 and 2 in Figure 3’s hard instances. In addition to removing singleton hyperedges, one could also consider adding the singleton hyperedge information as an extra vertex feature. However, the singleton hyperedges still need to be removed, ensuring that k-GWL can degenerate to k-WL for simple graphs, with consistent sub-hypergraphs and subgraphs.  We treat hyperedges as sets and do not allow duplicate vertices in hyperedges. But we allow repetitions of vertices in k-tuples, which is discussed in our response to Reviewer eKNU.
>
> >Q2. Regarding your result that k-folklore and (k+1)-oblivious WL are equally expressive: I understand that this is a proper generalization of the result obtained by Grohe & Otto for graphs. But your proof seems considerably simpler. In fact, Grohe & Otto requires heavy machinery based on linear algebra and linear programming to obtain this result. Please compare your result against them and explain how you managed to obtain the results without extending their machinery.
>
> Thank you for the good question. You are right that Grohe & Otto (2015) used heavy algebraic and linear programming methods to establish the equal expressivity between k-folklore and (k+1)-oblivious WL. Later, Grohe in his LICS'21 paper “The Logic of Graph Neural Networks” provided a considerably simpler less-than-one-page proof based on mathematical induction on iterations and analysis on the two variants’ computations. (See page 16 of https://arxiv.org/pdf/2104.14624, Proof of Theorem V.6 in Appendix.) We performed a careful adaptation of this simpler proof for hypergraphs, showing the equivalence of k-folklore and (k+1)-oblivious Generalized WL.

---

### Official Review · Reviewer_eKNU · 2025-03-13

**Overall Recommendation:** 4

**Summary:**

The paper defines a k-WL variant for hypergraphs which did not exist so far. The test relies on a new structure that only relies on the nodes of the hypergraph and follows the idea of k-FWL or k-WL (the oblivious variant) which differ in the order of aggregation. They additionally come up with a GNN variant of it which effectively generalizes k-WL from Morris et al 2019 (using the same tricks with sets instead of tuples and local aggregation, both making the algorithm theoretically weaker). For most levels, the difference between k-FWL and k-OWL for hypergraphs follows the same pattern as for graphs (with FWL being one level stronger), but for hypergraphs this does not hold on the first level, which is surprising.

**Claims And Evidence:**

All the claims are backed with evidence, typically proofs.

**Essential References Not Discussed:**

none.

**Experimental Designs Or Analyses:**

looks good (given that there are not many hypergraph datasets available and databases are too big).

**Methods And Evaluation Criteria:**

yes, proofing theorems about theoretical results indeed makes a lot of sense. Since there are not many hypergraph datasets, testing might be a bit harder. It would be interesting to see how the method works on databases (which are inherently hypergraphs), but those are typically too big to be handled by even 2-WL due to the exponential blowup.

**Other Comments Or Suggestions:**

Please note that the paper did NOT use the official template (but probably last year's template without the line numbers).

p3 first equation and eq 1 both have an extra pair of unnecessary brackets. Also eq 1 is not nicely indented (please align below the = symbol) same for eq 4 and 5 (and probably any multiline formula in the paper)

p3 the Babai 2016 citation is really unexpected here. It really has nothing to do with the whole paragraph (at least in my understanding)

p3 it would be nice to describe the exact difference between FWL and OWL here as it is used later on. Or at least do so in the appendix.

p3 end: please mention here that this compares tuples against sets and make explicit how duplicate entries in tuples are handled.

p3 def 4.1: I believe that is still just the atomic type that is known in logic. Or is there any difference (if yes, please mention it here)

p4 def of N(s): this looks to be identical to the graph setting, so it would be nice to state that this part is not new but rather standard.

p5: a good way to describe the difference between OWL and FWL is "all elements in one position" as opposed to "one element in all k positions".

p3 isomorphiosm -> isomorphism
p4 usng -> using
p6 typography in eq 6 and 7: please use \text{FWL} etc instead of just writing $GOWL$. This should also avoid the - sign to have that amount of space (in doubt, move it into the \text{} part.

**Other Strengths And Weaknesses:**

As mentioned before, I was struggling with lacking formality around sets, multisets, and tuples.

In section 3, please mention explicitly that hyperedges are unordered (which becomes relevant because the tuples of k-WL are ordered and k-WL somewhat expects relations to have an ordering as well). It would be nice if that difference was thoroughly discussed.

p5: it is only once mentioned indirectly how the hyperedges are included in the construction - namely only indirectly through the isomorphism type. This is not directly expected intuitively and could be mentioned explicitly. It took me quite a while to figure out that the whole construction really only depends on the vertex set and none of the introduced hyperedges depended on the actual hyperedges of the hypergraph we are working on. It would also be nice to mention that the runtime is independent of the number of hyperedges in the original hypergraph. The way it is written in sec 4.2 could be quite a bit clearer (as there is a set of hyperedges mentioned, but that does not seem to be the set of hyperedges of the original hypergraph, but rather of the constructed one).

p5: the examples are misleading as they do not consider repetitions which are (for standard k-WL) allowed.

Generally, I would have liked a very early explicit mention for the 2-GOWL vs 1-GFWL case instead of just writing this holds for k>=2. Because at least I was thinking about what that should mean and what is happening in the abovementioned case (as it is different from graphs).

sec 5.2: to me this section is not clear at all, what is done about permutations and how is the relation to individualization and refinement? (where you also put labels to stuff but in a way that does not change things). Furthermore, I would expect $\binom{n}{l}^l$ instead of without the exponent.

**Questions For Authors:**

My main questions are in the textbox about Other Strengths And Weaknesses. I am especially interested in the difference between sets, multisets, and tuples and would like the paper to be a bit more formal in that regard. Otherwise I really like the results (but the presentation could still be improved).


sec 7: is it common to exclude original vertex features? And how do results change if you add them?

Expressivity: injectivity and resulting equality in expressivity should only hold when its about tuples and not when its sets. Otherwise please write more about that difference here.

How small is the average hyperedge degree in Wri-Genre and why is that a problem? Is it really that different to the other datasets?

**Relation To Broader Scientific Literature:**

The paper very clearly states where it stands in term of hypergraph GNN research and also clearly shows the research gap.

**Theoretical Claims:**

I did not check the proofs, but the results are not surprising (except for the difference between 1-GFWL vs 2-GOWL which they showed using an example).

I would like to mention that the notation could be more clear and explicit in certain situations, especially when it comes to tuples, multisets, and sets and how they are interacting and how duplicates are handled.

---

> ### Author Rebuttal · Authors · 2025-03-30
>
> >As mentioned before, I was struggling with lacking formality around sets, multisets, and tuples. Misleading examples.
> Injectivity and resulting equality in expressivity should only hold when its about tuples.
>
> Thank you for the valuable comment. We should formally define tuples, multisets, and sets in k-GWL and will include the following definitions at the beginning of Section 4.1.
>
> “Tuples are ordered and allow repetitive elements; multisets are unordered and allow repetitive elements; and sets are unordered and do not allow repetitive elements. In k-GWL, we consider k-tuples, which allow repetitions of vertices. There are two types of hypergraphs: the input hypergraph and the k-tuple hypergraph (see its construction in Section 4.1.2). Hyperedges in both types of hypergraphs are sets, that is unordered.”
>
> We will add the following discussions on how to handle repetitions of vertices in k-tuples in Section 4. “Specifically, both k-GWL and k-WL allow repetitions of vertices in k-tuples. In the initialization of k-tuple features, they extract sub-hypergraphs and subgraphs induced by the *set* of vertices in k-tuples and then use the isomorphism type as features, where the set operation before extraction removes vertex repetitions. For the construction of k-tuple hypergraphs and k-tuple graphs, because tuples allow repetitions, the vertex replacement strategies in the oblivious and folklore variants ("all elements in one position" vs. "one element in all k positions") still work with no issues.”
>
> For better illustrating the construction of k-tuple hypergraphs, we will plot a new version of Figure 3 based on k-tuples, where repetitions of vertices are considered. For example, the folklore hypergraph now includes five hyperedges instead of only two. The hyperedge with $v_1$ in all three positions of $(v_1,v_2,v_3)$  includes $(v_1,v_2,v_3)$, $(v_1,v_1,v_3)$, and $(v_1,v_2,v_1)$. The oblivious hypergraph still has three hyperedges but $e_1$ with all vertices in the first position of $(v_1,v_2,v_3)$ includes two extra k-tuples $(v_2,v_2,v_3)$ and $(v_3,v_2,v_3)$.
>
> Last but not least, we will explicitly state that all our theoretical findings on the k-GWL hierarchy and equivalence of k-HNNs and k-GWL (theorems in Sections 5 and 6) are based on using k-tuples. The only place where we switch from k-tuples to k-sets is in the implementation of k-HNNs. Ignoring the vertex ordering and repetitions of vertices in k-tuples makes k-HNNs considerably more practical.
>
> >p5: it is only once mentioned indirectly how the hyperedges are included in the construction...
>
> Thanks for bringing this issue to our attention. In Section 4, we will emphasize that the construction of k-tuple hypergraph in k-GWL does not depend on the set of hyperedges in the original hypergraph, which is only used in the isomorphism type. In Section 6, we will point out, unlike k-GWL, for k-HNNs based on local neighborhoods, the actual hyperedges of the input hypergraph do influence the construction of the k-tuple hypergraph.
>
> >sec 5.2: what is done about permutations...
>
> Thank you for pointing out the issue. The original idea of relational pooling (that inspires $(k,l)$-WL) is to build expressive permutation-invariant models, where permutation invariance is obtained by averaging or summing over representations under all permutations of node IDs. Later on, local relational pooling performs permutation and averaging within a subgraph of size $l$ to reduce computational complexity. Similarly, $(k,l)$-WL assigns extra labels $1,2,\cdots,l$ to $l$ vertices, but runs k-WL on the whole graph with the additional labels only for those $l$ vertices. This is repeated for every possible labeled graph and then the final representation of the graph is aggregated from the representations of all labeled graphs. The extra vertex labels help boost expressivity. One can apply the same idea in k-GWL to get $(k,l)$-GWL. In the computational complexity, the term for the labelling should be $\binom{n}{l}^l$ (that is, select $l$ vertices from $n$ vertices and then assign $l$ labels to the $l$ vertices).
>
> >Other Comments Or Suggestions
>
> Thank you for your detailed comments. We sincerely appreciate your efforts in helping enhance the presentation of this manuscript and will apply these comments.
>
> >sec 7: is it common to exclude original vertex features? And how do results change if you add them?
>
> Keeping the original vertex features makes it easier to distinguish non-isomorphic hypergraphs, due to the potentially very different vertex features. Hence, it is a common practice to exclude these features and focus on structural information only.  We used the datasets from (Feng et al., 2024), which performed the pre-processing. Unfortunately, they have not provided the original vertex features, but we expect the accuracy would be higher with original vertex features being added.
>
> > Analysis of underperformance in Wri-Genre
>
> Please refer to the first response to Reviewer H4JM.

---

### Official Review · Reviewer_H4JM · 2025-03-14

**Overall Recommendation:** 3

**Summary:**

The paper introduces the k-dimensional Generalized Weisfeiler-Leman (k-GWL) algorithm, an extension of the classical Weisfeiler-Leman (WL) test to hypergraphs. The primary contribution is the formulation of k-GWL, which generalizes k-WL from graphs to hypergraphs, providing a unified theoretical framework for hypergraph isomorphism testing. Building on k-GWL, the authors introduce k-HNN (k-dimensional Hypergraph Neural Networks), which leverage k-GWL's structure to enhance hypergraph representation learning. Empirical evaluations on six real-world hypergraph classification datasets demonstrate that k-HNN achieves state-of-the-art (SOTA) performance.

**Claims And Evidence:**

Claims Supported:

- The expressivity hierarchy of k-GWL is rigorously proven (Theorems 5.1–5.4).

- The reduction of k-GWL to k-WL for graphs is well-supported by theoretical results.

- The claim that k-HNN outperforms existing hypergraph neural networks is empirically validated on diverse datasets.

**Essential References Not Discussed:**

It seems like [1] also discuss the expressivity of HNNs, but it is not cited by this paper.


[1] Luo, Zhezheng, et al. "On the expressiveness and generalization of hypergraph neural networks." arXiv preprint arXiv:2303.05490 (2023).

**Experimental Designs Or Analyses:**

Pro:
- Strong empirical performance across diverse datasets, confirming the practical benefits of k-HNN.

- Ablation studies (removal of singleton hyperedges) provide useful insights.

Con:
- No training time comparisons—it is unclear how much additional overhead k-HNN introduces compared to existing models.

- The underperformance on IMDB-Wri-Genre suggests that certain hypergraph structures may not benefit from k-GWL, requiring further investigation.

**Methods And Evaluation Criteria:**

- The chosen task of hypergraph classification aligns well with the proposed method.
- The benchmark datasets (IMDB, Steam-Player, Twitter-Friend) cover diverse domains, ensuring broad applicability.

**Other Comments Or Suggestions:**

- Comparison to Approximate WL Methods: How does k-GWL compare to randomized WL tests.
- Training Time Comparison: Can you provide training time vs. existing HNNs? How much overhead does k-GWL introduce?
- Can you explain more on the Definition 4.1? From my understanding, $s$ represents ID of nodes, then what does $s^1_{i_1}=s^1_{i_2}$ mean?

**Other Strengths And Weaknesses:**

Pro:
- Establishes a clear theoretical expressivity hierarchy for hypergraphs.

- Provides a unified framework that extends existing graph and hypergraph isomorphism methods.

Con:
- Computational overhead remains a concern, especially for large k.

- The method is only tested on classification tasks, limiting its generalizability.

**Questions For Authors:**

Same as comments.

**Relation To Broader Scientific Literature:**

- The paper builds on the Weisfeiler-Leman hierarchy and extends prior work on hypergraph neural networks.

- It contributes to hypergraph isomorphism testing by formalizing k-GWL as a higher-order method.

**Theoretical Claims:**

No, I did not check the proofs, but the overall statements make sense.

---

> ### Author Rebuttal · Authors · 2025-03-30
>
> >The underperformance on IMDB-Wri-Genre suggests that certain hypergraph structures may not benefit from k-GWL, requiring further investigation.
>
> Thank you very much for your comment. Following the suggestion of Reviewer yrE6, we have run experiments on k-HNNs for $k=3$ based on the vertex sampling strategy with sample size 10 in a GPU cluster with larger memory. As shown in the table below, 3-HNNs with the sampling show comparable performance to 2-HNNs across the datasets, while outperforming the previously best model AllSetTransformer in the Wri-Genre dataset. We will run the full 3-HNNs without the sampling (when there is no rebuttal time limit) and expect higher accuracy and include the results in the paper. We examined the dataset statistics in Table 5 of Appendix: there are six classes in Wri-Genre, which is significantly higher than in other datasets, making it a challenging dataset where every model seems to struggle. Moreover, in the two co-writer datasets (Wri-Genre and Wri-Type), there are significantly smaller number of hyperedges, with only about four hyperedges on average in a hypergraph, and each vertex participates in only 1.5 hyperedges on average. We was suspecting that the less amount of high-order information in the original hypergraph leads to a more dispersed construction of local neighbors between k-sets, which is not friendly to k-HNNs. However, the results of 3-HNNs with the sampling seem not support the thought.
>
> | Method               | IMDB_Dir_Form    | IMDB_Dir_Genre   | IMDB_Wri_Form   | IMDB_Wri_Genre  | Steam_Player    | Twitter_Friend  |
> |----------------------|------------------|------------------|-----------------|-----------------|-----------------|-----------------|
> | AllSetTransformer          | 66.76±2.55       | 79.31±0.94       | 52.67±4.77      | 54.09±2.41      | 61.87±1.87      | 64.03±1.34      |
> | 2-OHNN               | 67.25±2.35       | 79.75±1.14       | 55.35±3.74      | 50.08±1.89      | 65.97±1.37      | 64.12±1.30      |
> | 2-FHNN               | 68.11±2.46       | 78.52±1.07       | 55.36±3.30      | 45.44±1.24      | 67.53±1.23      | 62.75±2.18      |
> | 3-OHNN (Sample Size 10)   | 67.66±2.59       | 79.11±1.11       | 52.75±4.12      | 57.50±3.55      | 63.00±2.01      | 62.97±2.04      |
> | 3-FHNN (Sample Size 10)   | 67.07±2.33       | 78.90±1.1        | 50.56±2.69      | 53.84±2.05      | 61.50±1.59      | 63.05±3.67      |
>
> >It seems like [1] also discuss the expressivity of HNNs, but it is not cited by this paper.
>
> Thank you for sharing the paper. We will ensure the following discussions are included in the final version.
>
> “(Luo et al, 2022) studied the expressivity of hypergraph neural networks and constructed hierarchies of arity (the maximum number of vertices in hyperedges) and depth. For example, when the depth is larger than a certain value, a neural logic machine with a larger arity is more expressive. In contrast, this paper generalizes k-WL to k-GWL, unifies graph and hypergraph isomorphism tests, and establishes a clear expressivity hierarchy for hypergraphs.”
>
> >Comparison to Approximate WL Methods: How does k-GWL compare to randomized WL tests.
>
> We could not find papers introducing randomized WL tests after a Google search but do our best to discuss it based on our understanding. Our models with the vertex sampling become an approximation of the original k-GWL, effectively saving run-time at the expense of slight decrease in prediction accuracy. Randomized/approximate WL seems orthogonal with our k-GWL and can be combined for better computational efficiency.
>
> >No training time comparisons—it is unclear how much additional overhead k-HNN introduces compared to existing models.
> Training Time Comparison: Can you provide training time vs. existing HNNs? How much overhead does k-GWL introduce?
>
> Sorry for not making the results of training time and their comparison easier to see. In Table 8 of Appendix E.5, we report the average time per epoch for different hypergraph learning models. The run-times of our k-HNN methods are mostly smaller than double those of other compared methods. The vertex sampling approach can effectively reduce run-time to be closer to that of other methods. For both GNNs and HNNs, it is still an open problem on how to further improve the trade-off between expressive power and run-time.
>
> >Can you explain more on the Definition 4.1? From my understanding, $s$ represents ID of nodes, then what does $s^1_{i_1} = s^1_{i_2}$ mean?
>
> Sorry for the confusion. We use $s$ to represent a k-tuple and $s^1$ to represent a k-tuple in $HG_1$. The $i^{th}$ element in $s^1$ is referred to as $s_i^1$. $\forall i_1, i_2 \in [k], s^1_{i_1} = s^1_{i_2} \leftrightarrow s^2_{i_1} = s^2_{i_2}$ means that if the $i_1^{th}$ element and $i_2^{th}$ element in $s^1$ are the same, then the $i_1^{th}$ element and $i_2^{th}$ element in $s^2$ are also identical.

---

### Official Review · Reviewer_yrE6 · 2025-03-14

**Overall Recommendation:** 3

**Summary:**

This work generalize high-order weisfeiler-Lehman test and high-order GNNs on graph to hypergraph. Furthermore, it build expressivity hierarchy among different orders of WL test on hypergraph. Using it instead of hyperGNNs corresponding to 1-WL leading to performance increase on real-world datasets.

**Claims And Evidence:**

Yes. The proposed k-WL on hypergraph is meaningful for improving hypergraph network. The proposed method achieves provably high expressivity and better performance on real-world hypergraph datasets.

**Essential References Not Discussed:**

The related work is detailed and exhaustive, including representative expressive GNNs and representative HNNs.

**Experimental Designs Or Analyses:**

1. The number of parameters is not compared between baselines and the proposed models, which may lead to unfair comparison.

2. Models corresponding to k-WL with $k>2$ should also be included.

**Methods And Evaluation Criteria:**

Yes, the tasks and baselines are representative for hypergraph domain and can show the performance of hyperGNN clearly.

**Other Comments Or Suggestions:**

Text in figures are small and hard to read. Table 3 is too wide.

**Other Strengths And Weaknesses:**

The adaption of k-WL to hypergraph is too straightforward and lacks novelty. Improved complexity is also a weakness.

**Questions For Authors:**

Hypergraph can be bijectively mapping to a bipartite graph, with nodes and hyperedges in hypergraph as nodes in graph. Can we directly apply k-WL to this bipartite graph?

**Relation To Broader Scientific Literature:**

It is related to high-order WL test and high-order GNN commonly used in graph. This work adapts it to hypergraph. Some details like the initialization of k-WL on hypergraph is meaningful.

**Theoretical Claims:**

Yes. I checked the proof for main theorem 5.2 and 5.3. The language and proofsketchs are commonly used in expressivity analysis. The proof is correct.

---

> ### Author Rebuttal · Authors · 2025-03-30
>
> >The number of parameters is not compared between baselines and the proposed models, which may lead to unfair comparison.
>
> Thank you for the valuable comment. We have collected the number of parameters in all tested models and reported them in the table below. It can be observed that our k-HNN models use more than three times the parameters compared to several models, double the parameters of HNHN, but notably fewer parameters than AllSetTransformer. However, our proposed models and AllSetTransformer clearly outperform other baselines as shown in Table 1. For both graph and hypergraph deep learning, it is still an open problem on how to address the trade-off between expressive power and model complexity. In addition, we report the average time per epoch for all the models in Table 8 of Appendix E.5. The run-times of our methods are mostly smaller than double those of compared methods, while the vertex sampling approach can effectively reduce the run-time to be closer to that of other methods.
>
>
>
> | Models       | The number of parameters |
> |--------------|--------------------------|
> | MLP          | 20102                   |
> | HyperGCN     | 87942                   |
> | HNHN         | 187014                  |
> | UniGCNII     | 52230                   |
> | AllSetTransformer  | 390790                  |
> | ED-HNN        | 103558                  |
> | HIC          | 54406                   |
> | k-HNNs       | 365382                  |
>
> Note: k-HNNs have the same number of parameters for different values of k since they share the same neural network architecture.
>
> >Models corresponding to k-WL with k>2 should also be included.
>
> A: In our experiments, we had access only to a normal GPU server with a 1080Ti GPU with 11GB memory. Due to limited GPU memory, we considered only the case of $k = 2$. However, we recently obtained access to a GPU cluster with 40GB memory and have run experiments using our k-HNNs for $k=3$ based on the vertex sampling strategy with sample size 10. As shown in the table in our response to Reviewer H4JM, 3-HNNs with the sampling show comparable performance to 2-HNNs across the datasets, while outperforming the previously best model AllSetTransformer in the IMDB-Wri-Genre dataset. We will run the full 3-HNNs without the sampling (when there is no rebuttal time limit) and expect higher accuracy and include the results in the paper.
>
> >The adaption of k-WL to hypergraph is too straightforward and lacks novelty.
>
> k-WL cannot be directly applied to hypergraphs. The two main challenges in generalizing k-WL to hypergraphs are: (1) how to initialize the features of k-tuples through sub-hypergraph extractions while ensuring degeneration to k-WL, and (2) how to construct k-tuple hypergraphs from the original hypergraphs so that 1-GWL can be applied. Our key findings include: (a) establishing a Generalized WL hierarchy for hypergraphs with increasing expressivity, with the notable difference between 1-GFWL vs 2-GOWL, unlike its graph counterpart. (b) The hierarchy allows us to design hypergraph neural networks with the desired expressivity, improves upon most existing hypergraph neural networks whose expressive power is upper bounded by 1-GWL.
>
> >Text in figures are small and hard to read. Table 3 is too wide.
>
> A: Thank you for the comment to help improve the presentation of the paper. We will adjust the figures and table accordingly.
>
> >Hypergraph can be bijectively mapping to a bipartite graph, with nodes and hyperedges in hypergraph as nodes in graph. Can we directly apply k-WL to this bipartite graph?
>
> Thank you for the good question. One could transform hypergraphs to graphs via a bijective mapping, such as star expansion or line expansion [a], and then apply k-WL on the transformed graphs for isomorphism testing. However, even though the transformation is bijective, it does not guarantee the results of k-WL on the transformed graphs are the same as those of k-GWL on the original hypergraphs. For instance, we can find two non-isomorphic hypergraphs where k-GWL can distinguish them but k-WL cannot distinguish their transformed graphs. Figure 3(a) provides such an example: while 1-WL (equivalent to 2-OWL) fails to distinguish the transformed graphs, 2-GOWL successfully identifies the original hypergraphs. Therefore, this indirect method does not achieve the same expressive power as our k-GWL algorithm. We will include this observation in the final version.
>
> Reference:
>
> [a] Chaoqi Yang, Ruijie Wang, Shuochao Yao, and Tarek F. Abdelzaher. "Semi-supervised hypergraph node classification on hypergraph line expansion." In Proceedings of the 31st ACM International Conference on Information & Knowledge Management, pp. 2352-2361. 2022.

---

### Decision · Program_Chairs · 2025-05-01

**Decision:**

Accept (poster)

**Comment:**

The paper introduces a variant of the k-dimensional Weisfeiler-Leman test for hypergraphs, called k-GWL. The expressivity hierarchy of k-GWL is rigorously proven in Theorems 5.1–5.4. The paper additionally introduces a GNN variant of the k-GWL test (k-HNN) which generalizes k-WL from Morris et al 2019. Their k-HNN seems to outperform existing hypergraph neural networks empirically on some datasets.


The reviewers positively emphasized the strong empirical performance across diverse datasets, and the theoretical analysis of the expressivity hierarchy for hypergraphs introduced by the paper.


The reviewers also identified a few limitations:


1. Since there are not many hypergraph datasets, testing the method is challenging. It would have been more interesting to see how the method works on datasets which are inherently hypergraphs, but those are typically too big to be handled by even 2-WL due to the exponential blowup. This leads to the next point, which is a main limitation of the method: its computational overhead.


2. Computational overhead remains a concern, especially for large k.


3. In the experiments, the number of parameters is not the same between baselines and the proposed models, which makes it unclear in what sense the comparison is fair. Does the current paper use the recommended hyperparameters from the original papers that published each competing method? This is not written in the paper. If accepted, please add a section that discusses how the architecture and the hyperparameters were chosen for each competing method.


If accepted, please incorporate the reviewer’s comments in the camera-ready version of the paper.